# Effect of Adding *Bifidobacterium animalis* BZ25 on the Flavor, Functional Components and Biogenic Amines of Natto by *Bacillus subtilis* GUTU09

**DOI:** 10.3390/foods11172674

**Published:** 2022-09-02

**Authors:** Qifeng Zhang, Guangqun Lan, Xueyi Tian, Laping He, Cuiqin Li, Han Tao, Xuefeng Zeng, Xiao Wang

**Affiliations:** 1Key Laboratory of Agricultural and Animal Products Store & Processing of Guizhou Province, Guizhou University, Guiyang 550025, China; 2College of Liquor and Food Engineering, Guizhou University, Guiyang 550025, China; 3School of Chemistry and Chemical Engineering, Guizhou University, Guiyang 550025, China

**Keywords:** two-strain mixed fermentation, natto, flavor, free amino acids, biogenic amines, nattokinase

## Abstract

Natto is a high-value fermented soybean produced by *B. subtilis*. However, *B. subtilis* produces a pungent amine odor. This study compared the volatile organic compounds (VOCs), free amino acids (FAAs) and biogenic amines (BAs), nattokinase (NK) of natto made by two-strain fermentation with *Bifidobacterium animalis* BZ25 and *Bacillus subtilis* GUTU09 (NMBB) and that of natto made by single-strain fermentation with *Bacillus subtilis* GUTU09 (NMB). Compared with NMB, volatile amine substances disappeared, ketones and aldehydes of NMBB were reduced, and alcohols increased. Besides that, the taste activity value of other bitter amino acids was lowered, and BA content was decreased from 255.88 mg/kg to 238.35 mg/kg but increased NK activity from 143.89 FU/g to 151.05 FU/g. Correlation analysis showed that the addition of BZ25 reduced the correlation between GUTU09 and BAs from 0.878 to 0.808, and pH was changed from a positive correlation to a negative one. All these results showed that the quality of natto was improved by two-strain co-fermentation, which laid a foundation for its potential industrial application.

## 1. Introduction

Natto is a traditional soybean product fermented by *Bacillus subtilis* that contains various functional components, such as nattokinase (NK), vitamin K, isoflavones, superoxide dismutase, phospholipids, saponins, and linoleic acid [1,2]. Natto is a low-cost but high-nutrition product made using a simple procedure. Presoaked soybeans are cooked until tender, drained, cooled to 40 °C, inoculated with *Bacillus* natto, and incubated at 30–40 °C for 12–36 h. The production process of fermented soybeans mainly includes selecting soybeans, soaking, sterilization, steaming, inoculation fermentation, and after-ripening [3,4]. The active substances in natto exert preventive effects on thrombosis, high blood cholesterol, and cancer [5,6]. The health function of natto has always been the interest of consumers. Past research focused on extracting and increasing the functional components of natto, and there were few reports on improving the flavor and taste of natto. The flavor of natto is related to volatile compounds, and amines are the main cause of the unpleasant smell of natto [7].

Given that *Bifidobacterium animalis* BZ25 may produce lactic acid and acetic acid that mask amines [8,9], adding *B. animalis* may improve the VOCs of natto. The advantage of two-strain mixed fermentation could be attributed to the synergistic effect between different strains that improved the flavor of natto [10]. Hu et al. [11] found mixed *H. uvarum*/*S. cerevisiae* fermentation leads to the enhancement of wine’s fruity aroma. Wang et al. [12] found that the co-fermentation of *Saccharomyces Boulardii* and *Lactiplantibacillus plantarum* improved and modulated the flavor of green tea. In general, mixed fermentation/multi-strain starters are considered to have advantages over a single strain contributing to complexity, taste, and flavor [13]. Therefore, adjusting the formula of the start recipe of natto can enhance the sensory characteristics and general acceptability of the product and improve the content and activity of nattokinase. Currently, how *Bacillus subtilis* single bacteria and multiple strains affect the characteristics of natto is unclear [14]. Determining such an effect helps understand the role of strains in soybean fermentation for the preparation of high-quality natto.

Free amino acids (FAAs) are organic compounds that comprise basic amino and acidic carboxyl groups and exist only in a free state in organisms. Protein hydrolase degrades soybean protein to produce FAAs during soybean fermentation. FAAs can be divided into four categories by their sensory properties: the monosodium glutamate-like FAAs (Asp ad Glu), the sweet FAAs (Met, Ala, Gly, Ser, Pro, and Thr), the bitter FAAs (Arg, His, Ile, Leu, Phe, Trp, Lys, and Val), Astringent FAAs (Tyr) and the tasteless (Cys) [15]. The bitter amino acids in natto greatly affect the taste of natto, giving it a light, bitter taste. Excessively high contents of bitter amino acids worsen taste [16]. The composition of FAAs can be modified by using different food-processing technologies [17]. FAAs are one of the key quality indicators of natto taste, and FAA content affects the quality of final food products [18]. Therefore, this study investigated whether changing free amino acid content could improve natto taste through two-bacterium co-fermentation.

Biogenic amines (BAs) are a general term for a low-molecular-weight organic compound with biological activities and amino groups. They are important organic bases and participate in the metabolism of microorganisms. They are formed by the decarboxylation of FAAs, the amination of aldehydes and ketones, or the conversion of ammonia [19]. Histamine and tyramine are considered toxic. High histamine concentrations in plasma may lead to food poisoning that is characterized by diarrhea, dyspnea, or hypotension [20]. Spermine is involved in regulating cell growth and proliferation, as well as DNA transcription, RNA translation, protein biosynthesis, and immune response [21,22]. Given that natto products must be safe, their BA levels must be monitored.

Nattokinase (NK), a serine protease with strong fibrinolytic activity, is a serine protease that is composed of 275 amino acids and is produced by *B. subtilis* in natto [23]. NK can directly digest fibrin in blood vessels and is considered to be a promising drug for preventing and treating cardiovascular diseases [24]. NK activity is another important quality index of natto.

The present work, firstly, utilized mixed starters of *Bacillus subtilis* GUTU09 and *Bifidobacterium animalis* BZ25 to improve the flavor and taste of natto. The volatile substances in natto were analyzed by coupling headspace solid-phase microextraction (HS-SPME) and gas chromatography-mass spectrometry (GC-MS). HPLC was utilized to detect free amino acids to investigate the change of bitter amino acid content in natto. Secondly, NK and BA changes during fermentation were investigated to explore the feasibility of adding probiotic BZ25 to improve the natto’s flavor and functional components.

## 2. Materials and Methods

### 2.1. Materials

Soybeans were from Heilongjiang Province, China. The standards of 17 FAAs (His, Tyr, Arg, Val, Met, Phe, Ile, Leu, Ser, Gly, Ala, Thr, Lys, Asp, Glu, Cys, Pro) and eight BAs (putrescine, cadaverine, tyramine, spermidine, spermine, histamine, and 2-phenylethylamine), tetrahydrofuran, triethylamine, thrombin, and fibrinogen were bought from Solarbio (Beijing, China). Hydrochloric acid, sodium acetate, trichloroacetic acid, sodium bicarbonate, dansyl chloride, perchloric acid, sodium hydroxide, Tris-HCl, and ammonia were procured from Sinopharm Chemical Reagent Co., Ltd., Shanghai, China. All the above reagents were of analytical grade. 2-Methyl-3-heptanone was of chromatographic grade and bought from Solarbio (Beijing, China). Methanol and acetonitrile were of chromatographic grade and purchased from Beijing Bailingwei Technology Co., Ltd., Beijing, China.

### 2.2. Preparation of Natto

#### Seed Culture

The two strains were screened by our lab. *Bifidobacterium animalis* subsp. *lactis* BZ25 was inoculated into De Man, Rogosa, Sharpe (MRS) liquid medium plates [25] and cultured at 37 °C for 24–48 h in a constant-temperature CO_2_ incubator. *B. subtilis* GUTU09 was inoculated into the seed culture medium (containing 10 g/L of glucose, 5 g/L of yeast extract, 10 g/L of beef extract, and 5 g/L of NaCl with pH 7.0–7.5). The temperature was set at 37 °C for 18 h, and the cells were harvested, pelleted, and resuspended in sterilized physiological saline at 1.0 × 10^8^ CFU/mL for inoculation. *Bacillus subtilis* GUTU09 was isolated from the traditional fermented soybeans in Guizhou Province, China. BZ25 was from Guizhou Xiang Pigs. *B. subtilis* GUTU09 was preserved in the CCTCC (CCTCC M 2021641), and *Bifidobacterium animalis* subsp. *lactis* BZ25 was preserved in the CGMCC (CGMCC NO. 10225).

Intact and uniform soybeans were soaked in deionized water for 12–14 h. After water filtration, 50 g of soybeans was placed into a triangular flask, added with 1% sucrose, and sterilized. Single-strain fermentation was performed with 3.5% of inoculum volume. Two-strain mixed fermentation was performed with the 1:1 ratio of BZ25 to GUTU09 and a 7% of inoculum volume at 35.5 °C for 30 h. In this work, steamed soybean (SS) samples were obtained after the sterilization of soaked soybeans. Steamed soybean (SS), soybean fermented by BZ25 (SFB), and natto made by GUTU09 (NMB) were used as the controls, and natto made by two mixed strains samples (NMBB) were made and analyzed.

### 2.3. Determination of Total Viable Count (TVC)

One gram of natto was homogenized for 15 s in 9 mL of sterilized physiological saline, serially diluted, and poured onto MRS medium plates or Luria-Bertani (LB) plates (10 g/L of tryptone, 5 g/L of yeast extract, 10 g/L of NaCl and 20 g/L of gram agar) to assess the TVC. *B. animalis* BZ25 was cultured anaerobically in MRS plates at 37 °C for 48 h, and *B. subtilis* GUTU09 was cultivated aerobically in LB plates at 37 °C for 24 h. The colonies were counted, and the viable counts were expressed as colony-forming units per gram of sample (CFU/g).

### 2.4. Determination of Titratable Acidity (TA) and the pH Value

Titratable acidity was measured by titration with 0.01 M sodium hydroxide based on the national standards method GB/T 12456-2021. It was expressed in the content of lactic acid, and the conversion factor was 0.09. The pH of fermented soybean was measured using a pH meter (PHSJ-3F, Chengdu Century Ark Technology Co., Ltd., Chengdu, China).

### 2.5. Solid-Phase Microextraction–GC–MS Analysis

#### 2.5.1. Sample Treatment

Samples were treated through solid-phase microextraction (SPME). Three grams of sample were placed in a 20 mL-headspace bottle. One μL of 2-methyl-3-heptanone (0.525 μg/mL) was then added to the sample. An aged 50/30 µm CAR/PDMS/DVB extractor (Supelco, Bellefonte, PA, USA) was inserted into the headspace of the sample bottle and adsorbed at 60 °C for 30 min. The adsorbed extractor was removed and inserted into the GC inlet and desorbed at 250 °C for 3 min. At the same time, the instrument was started for data collection.

#### 2.5.2. Separation of Volatiles

Samples were separated and analyzed via GC–MS (GC, 7890A, Agilent Technologies Ltd., Santa Clara, CA, USA; MS, 5975C, Agilent Technologies Co., Ltd., Multifunction Automatic Sampler, PAL/RSI, Stes Analytical Instruments, Villaz-St-Pierre, Switzerland) equipped with a DB-wax column (30 m × 0.25 mm × 0.25 µm). The heating program was as follows: the initial temperature was 40 °C and maintained for 3 min, then increased to 230 °C at a rate of 10 °C/min and retained for 5 min. The flow rate was 1 mL/min. The MS conditions included an emission current of 1 mA, electron energy of 70 eV, interface temperature of 250 °C, emission source temperature of 200 °C, and detector voltage of 2000 V.

#### 2.5.3. Qualitative and Quantitative Analyses of Samples

Referring to the method of Tian et al. [26], the qualitative methods for the determination of volatile flavor compounds in natto included MS library retrieval for MS and retention index (RI) calculation. Volatile flavor compounds were identified by using the Nist 2.0 database. The volatile compound concentration in the natto sample was calculated based on the relationship between the peak areas of the internal standard solution and the peak area of the sample compound.

#### 2.5.4. Odor Activity Value (OAV)

Odor activity value (OAV) was calculated as follows:OAVi = Ci/OTi

Ci represents the content of the volatile compound detected in the sample. OTi represents the odor threshold of this compound found in the literature. The volatile compound with OAV equal to or greater than one contributes to the aroma as an odor-active compound because its content is above its odor threshold, whereas those with OAV smaller than one may not.

### 2.6. Sensory Properties

The sensory characteristics of natto were evaluated using the procedure by Yang et al. [27]. A sensory team consisted of 20 teachers and graduate students majoring in food-related courses with sensory evaluation experience. The samples were labeled with different numbers and then randomly disrupted. The sensory characteristics of natto were mainly determined by its appearance, stickiness, flavor, taste, and chewiness. The appearance of natto was evaluated by darkness and uniformity. The stickiness was measured by how well natto clung to chopsticks. The flavor of natto was evaluated by olfaction, ammonia odor, and bean odor. Good natto should have a plain or somewhat sour smell and taste, whereas an ammonium odour and a dead leaf odour are undesirable. The taste was expected to be slightly acidic or mellow. The chewiness is the feedback of teeth on the softness, hardness, stickiness, and smoothness of natto. For all the sensory traits, rating scores from 1 to 5 were used, where 5 = excellent, 4 = good, 3 = moderate, 2 = poor, and 1 = inferior. Finally, a high index score indicated good natto quality.

### 2.7. Determination of FAA Content

The method used for FAA content in this work was consistent with that used by Liyanaarachchi et al. [28]. The soybean samples were pre-frozen at −20 °C for 12 h, then transferred to a vacuum freeze dryer (GTFD-12 S, Beijing Yonghe Chuangxin Electronic Technology Co., Ltd., Beijing, China) for freeze-drying for 48 h, then removed and ground into fine powder. A total of 1.00 g of the powder sample was treated with 25 mL of 5% trichloroacetic acid. After ultrasonic treatment at room temperature for 20 min, the sample was allowed to stand for 3 h. After filtration with double-layered filter paper, 4 mL of the filtrate was collected and centrifuged at 8000× *g* in a 5 mL-centrifuge tube for 30 min (Shanghai Anting Scientific Instrument, Shanghai, China). After the sample was filtered again with 0.22 µm water film, it was determined by HPLC (Agilent 1100, Agilent Technologies Co., Ltd., Santa Clara, CA, USA).

The chromatographic column was Agilent Hypersil ODS (5 µm, 250 mm × 4.0 mm, Agilent Technologies Co., Ltd., Santa Clara, CA, USA). UV detection wavelength of all amino acids was 338 nm, except that proline was detected at 262 nm. Mobile phase A (pH = 7.2) was 27.6 mmol/L sodium acetate: triethylamine: tetrahydrofuran (volume ratio 500:0.11:2.5). Mobile phase B (pH = 7.2) was 80.9 mmol/L sodium acetate: methanol: acetonitrile (volume ratio 1:2:2), and the flow rate was 1.0 mL/min. The column temperature was 40 °C, and the contents of amino acids were determined using external standards. Gradient conditions were as follows: 0 min–17 min, 92% A–50% A; 17 min–20.1 min, 50% A–0% A; 20.1 min–24 min, 0% A–92% A. According to the taste characteristics of amino acids, taste activity value (TAV) was analyzed and calculated. TAV is the ratio of amino acid content to its taste threshold.

### 2.8. Determination of NK Activity

Our previous report collected the sample supernatant to analyze the NK activity (FU/g) of fermented soybeans through the fibrin degradation method [29]. Ten grams of fermented soybeans were dissolved in 90 mL deionized water and homogenized in a beating machine for 10 s, then extracted at 4 °C for 24 h, and centrifuged at 10,000× *g* for 10 min. The specific steps were as follows: 1.4 mL of Tris-HCl (50 mM, pH 7.8) buffer and 0.4 mL of fibrinogen solution (7.2 mg/mL) were added to a test tube in a 37 °C water bath for 5 min, followed by adding 0.1 mL of thrombin (20 U/mL) and then bathed in water at 37 °C for 10 min to form artificial thrombosis. Next, adding 0.1 mL of the sample supernatant to the tube, it was continued to bathe in water at 37 °C for 60 min, and then the reaction was terminated by adding 2 mL of trichloroacetic acid (0.2 moL/L) solution standing for 20 min. Subsequently, the reaction solution was centrifuged at 10,000× *g* for 10 min, and the supernatant was analyzed at 275 nm wavelength by a spectrophotometer (UV-2700, Shimadzu Enterprise Management Co., Ltd., Kyoto, Japan). Definition of enzyme activity: the amount of enzyme required for an increase of 0.01 absorbance at 275 nm per minute is defined as one FU of fibrin degradation enzyme activity.

### 2.9. Determination of BAs

The sample treatment method used in this work was consistent with the report by Kim et al. [30]. The specific steps were as follows: the fermented natto was ground evenly in a bowl, and 5.00 g of the sample was placed in a 50 mL centrifuge tube. Then, 20 mL of 0.4 M perchloric acid was added. The mixture was vortex mixed until uniform and subjected to oscillating extraction for 30 min, and then centrifuged at 8000× *g* for 10 min. The supernatant was used for standby, and the same volume of 0.4 M perchloric acid was used to extract the residue again. Then two supernatants were combined, and the final volume was adjusted to 50 mL with 0.4 M perchloric acid.

BA content was measured by the method of Li et al. [31]. One mL of sample solution was placed in a 10 mL- centrifuge tube. Then, 200 µL of sodium hydroxide solution (2 mol/L) and 300 µL of saturated sodium bicarbonate solution were added successively. Subsequently, two mL of dansyl chloride derivative (10 mg/mL with acetone as solvent) was added. After shaking and mixing, the mixture was bathed in water at 40 °C for 45 min under dark conditions and oscillated once for 15 min. The derivative reaction was terminated by adding 100 µL of 25% ammonia water, and the sample was kept in the dark for 30 min. The sample was mixed with 1400 µL of pure acetonitrile, filtered with a 0.22 µm organic filter, and stored in the dark at 4 °C for HPLC analysis. The chromatographic column was 4 µm Nova-Pak C18 (150 mm × 3.9 mm, Waters). The injection volume was 20 µL, the column temperature was 30 °C, and the UV detection wavelength was 254 nm. Mobile phase A was chromatographically pure acetonitrile. Mobile phase B was deionized water. The flow rate was 0.8 mL/min. Gradient conditions were as follows: 0 min–15 min, 60% A–70% A; 15 min–22 min, 75% A–85% A; 22 min–25 min, 85% A–90% A; 25 min–28 min, 90% A–90% A; 28 min–35 min, 90% A–60% A.

### 2.10. Statistics and Analysis

Three parallel tests were performed for each sample, and the results were expressed as mean ± standard deviation (SD). Analysis of variance (ANOVA) with Duncan’s multiple range test was applied to evaluate data significance (*p* < 0.05). SPSS 20 software (SPSS Inc., Chicago, IL, USA) was used for data analysis, and Origin 2019 (Origin, Inc., Princeton, NJ, USA) was used to plot charts. Simca 14.1 was used for principal component analysis (PCA) plotting, and flavor heatmap was performed in the R environment (https://www.r-project.org/, accessed on 15 June 2022) with heatmap packages. The heat maps of correlation analysis were plotted using Chiplot (https://www.chiplot.online/, accessed on 19 June 2022).

## 3. Results and Discussion

### 3.1. TVC of Soybean Fermented by BZ25 (SFB), Natto Made by GUTU09 (NMB), or Natto Made by Two Mixed Strains (NMBB)

Viable count (VC) is an important indicator of natto product. Figure 1A shows the effect of fermentation time on the viable cells of GUTU09 and BZ25. After 12 h fermentation, VC of BZ25 in NMBB was lower than in NMB. All GUTU09 grew rapidly at the early fermentation stages, and their growth rate gradually decreased as fermentation progressed. VC of BZ25 was always lower than that of GUTU09 in NMBB. Fermentation was done in an aerobic environment, and BZ25 on the surface of natto was affected by oxygen, stunting its growth. After two-strain fermentation for 30 h, the average maximum VC of GUTU09 and BZ25 were 9.81 and 8.8 log CFU/g, respectively. TVC of NMB was 11.58 log CFU/g. Yoon et al. [32] found that the growth of *Bacillus subtilis* was inhibited during co-fermentation, and this could be caused by acid produced by *Lactobacillus plantarum*.

### 3.2. pH and Titratable Acid (TA) Analysis

As shown in Figure 1B,C, after 9 h, TA content in the NMB sample decreased, while that in the NMBB continued to increase. The addition of BZ25 could increase acid content in the sample and decrease pH. The TA content of NMBB was 1.494 ± 0.054 g/kg, and that of NMB was 0.696 ± 0.01 g/kg. The TA content of NMBB was higher than that of NMB by 0.798 g/kg. From 21 h, titratable acid in NMBB was significantly higher than that in SFB. Results showed that the co-fermentation of two strains has a synergistic effect on acid production, which promoted the acid production of BZ25. Bujna et al. [33] found that more acids were produced by co-fermentation of *Lactobacillus* and *Bifidobacterium*. The pH of NMB samples showed an upward trend, while the pH of SFB and NMB samples continued to decrease, indicating that BZ25 has a strong acid-producing capacity and changes the acidic environment of natto, thereby affecting the flavor of natto. Li et al. [34] found that the addition of *bifidobacteria* would reduce the pH of the milk fermented by *lactobacillus plantarum*, which was consistent with the results from this study.

### 3.3. Effect of BZ25 Addition on the VOCs of Natto

Given that strain types and production processes can affect fermentation and then change the composition of aroma components. SS, SFB, NMB, and NMBB were identified by HS-SPME-GC-MS. As shown in Appendix A, 133 kinds of VOCs were identified. Pyrazines, ketones, and acids are abundant in NMB. The main VOCs in NMBB were pyrazines, ketones, acids, and alcohol. The difference between the main flavor substance in the NMB and the NMBB was alcohol, which was caused by the addition of BZ25. Compared with similar reports, it is found that the main VOCs of natto are the same but also different, reflecting both commonness and individuality. For example, Kaczmarska et al. [35] found the volatile profile of the natto was dominated by alkylpyrazines and ketones. Chen et al. [7] found that the main VOCs in natto were esters, ketones, pyrazines, and phenols. The different strains and fermentation conditions lead to the difference in natto volatiles components. To further examine the contribution of these 133 volatile compounds to the overall aroma profile of natto, their OAVs were calculated according to their concentrations and odor thresholds (Appendix A).

As shown in Table 1, VOCs with OAV > 10 in Appendix A are listed separately to more clearly find the substances that contribute significantly to the overall flavor of the sample. Trimethyl-pyrazine, 2,3-butanedione, decanal, nonanal, 1-octen-3-ol, acetic acid, and 2-methoxy-phenol were the main VOCs in NMBB. Pyrazines were an important type of VOCs in natto. Pyrazines are formed by *Bacillus subtilis* or *natto* bacteria [36]. Liu et al. [1] found that the key VOCs in natto were 2,3-butanedione, 5-methyl-2-hexanone, furaldehyde, acetic acid, and 2,5-dimethylpyrazine, etc. Ketones can be generated by amino acids or by the Maillard reaction [1]. Kimura and Kubo [37] found that the key VOCs in natto were 2,3-butanediol, geranylacetone, isobutyl acetate, isovarelic acid, 2,5-dimethylpyrazine, and trimethylpyrazine, etc. The key VOCs in the NMB were benzeneacetic acid methyl ester, 2-ethyl-3,5-dimethyl-pyrazine, 2-heptanone, acetic acid, 2-ethyl-butanoic acid, 2-methyl-butanal, 2-ethyl-furan, and 2-methoxy-phenol. According to the VOCs of NMB, NMBB, and the natto reported in these articles, it was found that pyrazines were the main VOCs of natto, which indicated that *Bacillus subtilis* or *natto* bacteria could synthesize pyrazines during the fermentation of natto. The differences in VOCs also reflect personality differences. Most amines have a stimulating taste, which was the main reason for the ammonia flavor of natto. *N*,*N*-dimethyl-methylamine, formamide, 3-methyl-butanamide, and ethosuximide were detected in NMB but not in NMBB. Natto fermented by a single bacterium produces an ammonia smell, which is consistent with the report from Chen et al. [38]. The disappearance of volatile amine indicated that the ammonia taste of natto was effectively reduced. After the addition of BZ25, volatile amine disappeared, volatile aldehydes and ketones species decreased, and volatile alcohol increased. Alcohols may be produced from the metabolism of unsaturated fatty acids of microorganisms [39]. The VOCs of natto were usually related to microbial metabolism [40]. Compared with the sensory evaluation results, it is the disappearance of some irritating volatile substances that makes the NMBB sample more popular. Sadineni et al. [41] found that the co-fermentation of two strains during mango wine fermentation may decrease volatile acid and improve flavor.

#### Principal Component Analysis (PCA) and Hierarchical Cluster Analysis (HCA)

To visualize a total picture of the distributions of 73 odor-active compounds (OAV > 1) in all samples, PCA was applied (Figure 2A). PCA showed the major differences among the volatile profiles of four samples. The first principal component (PC1) of the PCA accounted for 50.1%, and PC2 explained an additional 29.7% of the total variability. And the cumulative contribution rate reached 79.8%. PCA Score Plot (Figure 2A) shows that NMB and SFB were located in the first and second quadrants and positively correlated with PC2. SS and NMBB were located in the third and fourth quadrants and negatively correlated with PC2. Four samples were distributed in different quadrants, indicating that there were significant differences in the flavor substances between samples. The characteristic components of SS were 2-methyl-propanoic acid ethyl ester (B4), 3-octanone (D8), 3-methyl-butanal (F3), 6-methyl-5-hepten-2-one (D19), phenylacetaldehyde (F6), eucalyptol (G20), 2-pentyl-furan (I1), 2-ethyl-furan (I3); those of SFB were 1-hexanol (G4), 4-methyl-3-penten-2-one (D15), and 2-methoxy-phenol (H1); those of NMB was *N*,*N*-dimethyl-methylamine (A1), 2-ethyl-3,5-dimethyl-pyrazine (C3), acetic acid (E1), 2-methyl-butanal (F4), 2-methyl-3-hexanol (G9), and 2-methyl-butanal (F4), and those of NMBB were hexanoic acid ethyl ester (B1), trimethyl-pyrazine (C1), 2-ethenyl-6-methyl-pyrazine (C8), 2-hexadecanone (D3), 2,3-butanedione (D4), nonanal (F2), 3-octanol (G3), biphenyl (H2), styrene (J3), and naphthalene (J6). Therefore, the characteristic flavor components of natto can be changed by mixed fermentation of two bacteria.

Figure 2B shows hierarchical clustering and heat map visualization of volatile compounds of four samples in different treatments (OAV > 10). The gallery plot of four samples was applied to visualize the differences in aroma compounds. The aroma components of natto produced by co-fermentation were different from those produced by single strain GUTU09. Next, we used clustering analysis for volatile compounds content and found that the four groups could be divided into three categories. Class I (SFB and NMB) included high content of acetic acid (E1), 2-ethyl-3,5-dimethyl-pyrazine (C3), 2-methyl-butanal (F4), 1-hexanol (G4), 2-methyl-3-hexanol (G9) and 2-methoxy-phenol (H1). Class II (SS) contained high content of 2-pentyl-furan (I1), 2-ethyl-furan (I3), hexanal (F9), 6-methyl-5-hepten-2-one (D19), eucalyptol (G20), 3-methyl-butanal (F3), 2-methyl-propanoic acid ethyl ester (B4), octanal (F5), 3-octanone (D8) and phenylacetaldehyde (F6). Class III (NMBB) included high content of 2,3-hexadecanone (D3), 2,5-dimethyl-pyrazine (C2), trimethyl-pyrazine (C1), hexanoic acid ethyl ester (B1), 1-octen-3-ol (G1), styrene (J3), biphenyl (H2), 3-octanol (G3), naphthalene (J6), 2,3-butanedione (D4) and 2,3-dihydro-benzofuran (I2). Figure 2B shows that the VOCs of NMBB accumulated more than that of NMB, and the flavor substances that contribute more to the whole are very different from NMB. PCA plot and gallery map show that mixed fermentation can greatly change the composition of VOCs of natto, thus affecting the sensory score and increasing people’s acceptance of natto.

### 3.4. Sensory Properties

The sensory properties of NMBB and NMB were evaluated in terms of appearance, stringiness, flavor, taste, and chewiness. As shown in Figure 3, there was no significant difference in appearance, stringiness, and chewiness scores between the two samples. The flavor of the NMBB was 3.98, while that of the NMB was 2.94. The taste score of the NMBB was 4.25, while that of the NMB was 3.6. NMBB was presented with cocoa, sour, and roasted nutty fragrance. The NMB showed ammonia, fruity, and cocoa fragrances. During fermentation, the amounts of ammonia are released to give a negative influence on the flavor of natto [38]. The disappearance of amine volatiles and the decrease of bitter amino acids make NMBB more popular. The increase in the flavor score was attributed to the lactic acid produced by BZ25, which neutralized amines and gave natto a slightly acidic odor. BZ25 plays a major role in reducing the ammonia taste and bitter taste of NMBB. The sensory properties of fermented soybean products are affected by various microorganisms, and microbial enzymes also influence the characteristic taste and flavor of products [50]. Soy protein can be hydrolyzed by proteases to produce peptides and amino acids. The bitter taste of natto during chewing may be related to bitter peptides and amino acids such as leucine, histidine, and proline, especially peptides and tyrosines produced by casein hydrolysis [51]. From the data of FAAs (Table 2), it can be seen that the mixed fermentation of two bacteria can reduce the bitter taste of natto. The decrease in bitter amino acid content is also one of the reasons for the increase in the taste of NMBB. The sensory evaluation results showed that mixed fermentation was beneficial in improving the flavor of natto.

### 3.5. Effect of BZ25 Addition on FAAs and Bitter Amino Acids

The FAA contents of four samples were detected and shown in Table 2. FAA content affects the quality of food products [18]. Bitter amino acids are rated as the leading influencing factor of poor taste [52]. FAAs in food, such as lysine, threonine, and methionine, affect the protein quality and flavor of food [53]. Eight bitter amino acids were detected in NMBB and NMB. Except for His, the TAV of bitter amino acids (NMBB) was lower than that of the NMB. In both groups of samples, Lys had the highest TAV, which was 7.0 and 9.9, respectively. In natto, the acid produced by BZ25 reduced pH. It also decreased the capability of protease to hydrolyze protein into FAAs by inhibiting protease activity. The content of FAAs can reflect the degree of protein degradation in soybean samples [54]. Proteases were produced during the growth and metabolism of GUTU09. These enzymes promoted the hydrolysis of proteins into various peptides, such as short peptides and amino acids. These substances could be decomposed and oxidized by BZ25 and used as nitrogen sources for bacterial growth, which led to the reduction in FAA content. This effect could be another reason for the reduction in FAA content. The above data showed that two-strain mixed fermentation could decrease the content of bitter amino acids.

**Table 2 foods-11-02674-t002:** Content of free amino acids and taste activity value (TAV) in SS, SFB, NMB and NMBB samples.

Name	RT (min)	Free Amino Acids Content (mg/g)				Threshold Value (mg/g)	Taste Activity Value (TAV)			
		SS	SFB	NMB	NMBB		SS	SFB	NMB	NMBB
His	7.32	0.08 ± 0.02 ^c^	0.4 ± 0.1 ^b^	0.52 ± 0.14 ^ab^	0.58 ± 0.06 ^a^	0.19	<1	2.1	2.7	3.1
Tyr	12.88	0.27 ± 0.07 ^c^	0.3 ± 0.04 ^c^	3.16 ± 0.17 ^a^	2.51 ± 0.12 ^b^	2.6	<1	<1	1.22	<1
Arg	9.95	3.7 ± 0.18 ^b^	4.34 ± 0.13 ^a^	0.45 ± 0.11 ^c^	0.36 ± 0.15 ^c^	0.5	7.40	8.68	<1	<1
Val	17.23	0.17 ± 0.02 ^c^	0.15 ± 0.03 ^c^	2.3 ± 0.08 ^a^	1.75 ± 0.06 ^b^	0.487	<1	<1	4.7	3.6
Phe	20.14	0.24 ± 0.02 ^c^	0.11 ± 0.08 ^c^	4.1 ± 0.3 ^a^	3.71 ± 0.03 ^b^	1.092	<1	<1	3.8	3.4
Ile	20.54	0.11 ± 0.02 ^c^	0 ^c^	1.84 ± 0.25 ^a^	1.22 ± 0.12 ^b^	0.9	<1	0	2.0	1.4
Leu	21.9	0.18 ± 0.03 ^c^	0.04 ± 0.02 ^c^	5.08 ± 0.36 ^a^	3.38 ± 0.15 ^b^	0.846	<1	<1	6.0	4.0
Lys	22.54	0.18 ± 0.07 ^c^	0.21 ± 0.12 ^c^	4.95 ± 0.25 ^a^	3.5 ± 0.08 ^b^	0.5	<1	<1	9.9	7.0
Ser	6.37	0.01 ± 0.01 ^a^	0.04 ± 0.05 ^a^	0.04 ± 0.01 ^a^	0.33 ± 0.48 ^a^	2.2	<1	<1	<1	<1
Gly	8.28	0.11 ± 0.02 ^c^	0.09 ± 0.08 ^c^	1.74 ± 0.1 ^a^	0.87 ± 0.13 ^b^	1.30	<1	<1	1.3	<1
Ala	10.83	0.41 ± 0.04 ^c^	0.24 ± 0.06 ^c^	2.36 ± 0.18 ^a^	1.43 ± 0.12 ^b^	1.44	<1	<1	1.6	<1
Thr	8.6	0.2 ± 0.3 ^bc^	0.08 ± 0.05 ^c^	0.62 ± 0.1 ^a^	0.47 ± 0.02 ^ab^	3.061	<1	<1	<1	<1
Met	17.51	0.13 ± 0.01 ^c^	0.06 ± 0.04 ^c^	1.03 ± 0.17 ^a^	0.8 ± 0.04 ^b^	0.555	<1	<1	1.90	1.40
Pro	29.45	0.31 ± 0.09 ^b^	0.24 ± 0.03 ^b^	1.57 ± 0.2 ^a^	1.37 ± 0.05 ^a^	1.738	<1	<1	<1	<1
Asp	2.91	0.24 ± 0.07 ^c^	0.18 ± 0.08 ^c^	1.25 ± 0.17 ^a^	0.93 ± 0.02 ^b^	0.216	1.1	<1	5.8	4.3
Glu	3.26	0.77 ± 0.05 ^c^	0.12 ± 0.08 ^d^	5.5 ± 0.21 ^a^	4.23 ± 0.15 ^b^	1.437	<1	<1	3.8	2.9
Cys	15.64	0.01 ± 0.01 ^c^	0.03 ± 0.02 ^c^	0.45 ± 0.02 ^a^	0.24 ± 0.13 ^b^	n.f.	—	—	—	—

Note: RT: Retention Time; SS: steamed soybean; SFB: soybean fermented by BZ25; NMB: natto made by GUTU09; NMBB: natto made by two mixed strains; the monosodium glutamate-like FAAs (Asp ad Glu), the sweet FAAs (Met, Ala, Gly, Ser, Pro, and Thr), the bitter FAAs (Arg, His, Ile, Leu, Phe, Trp, Lys and Val), Astringent FAAs (Tyr) and the tasteless (Cys); Values are mean ± SD (*n* = 3); Means with different lower case letters (a–d) in the same row indicate a significant difference (*p* < 0.05); Thresholds taken from Refs. [55,56]. Odor description found in the literature (Flavornet); “n.f.”, “—”: Data was not found in literatures.

### 3.6. Effect of BZ25 Addition on NK Activity

As shown in Figure 4, a significant difference was observed in the changes in NK activity (*p* < 0.05). NK activity gradually increased with the extension of fermentation time. Before 18 h, The NK activity in NMB was higher than that in the NMBB sample. The NK activities in both groups of samples were similar at 18 h. After more than 18 h of fermentation, NMBB showed higher NK activity than NMB. From Figure 1, the number of viable cells of GUTU09 in NMB is much higher than that of NMBB, and it is obvious that NK is produced by GUTU09. That indicates that it is not that the more viable bacteria GUTU09 is, the higher its NK-producing activity is, and other factors, such as BZ25, regulate its NK-producing. There are many cases of two bacteria co-fermentation promoting the production of metabolites. For example, Chen et al. [57] found that the total phenolic and genistein contents were increased by co-fermentation of *Lactobacillus* with *Bacillus subtilis*. In the later stage of fermentation, BZ25 promoted the production of NK by GUTU09. NK increased from 143.89 FU/g to 151.05 FU/g after 30 h of fermentation. In natto, the addition of BZ25 promoted the accumulation of NK and the increment in the content and quality of active substances of natto.

### 3.7. Effect of BZ25 Addition on Bas

#### 3.7.1. Changes in Bas during Fermentation

As shown in Figure 5A,B, only five Bas including putrescine, cadaverine, tyramine, spermidine, and spermine, were detected in NMB and NMBB. A significant difference (*p* < 0.05) was observed in the changes in spermine and tyramine contents (Figure 5). Cadaverine and putrescine have no direct toxicity to the human body and are considered natural amines in food [58]. Tyramine contents exceeding 100 mg/kg and β-phenylethylamine contents over 30 mg/kg are considered to be toxic doses [59]. Histamine and 2-phenylethylamine were not detected in all samples. The highest content of tyramine was 55.78 ± 1.79 mg/kg, which did not exceed limits, indicating that the strain used for fermentation had good biological safety.

The change curves of putrescine and cadaverine in the NMBB and NMB were relatively flat as fermentation progressed. The spermine content in the NMB showed an increasing trend, which in the NMBB sample decreased slightly at first and then increased continuously. From 0 h to 30 h, the spermine content in the NMBB increased from 21.12 ± 0.22 mg/kg to 119.65 ± 1.66 mg/kg. The minimum concentration of spermine required to exert cytotoxicity was 653.56 mg/kg [60]. The spermine content in NMBB was 119.65 mg/kg, which was far below the minimum toxic concentration. Bas are common in fermented soybean foods. High levels of Bas were found in 21 different natto in Korea [30], among which nine natto had β-phenylethylamine or tyramine contents higher than the toxic dose (30 mg/kg and 100 mg/kg, respectively) of each amine. However, the content of each BA in NMB and NMBB did not exceed its limit, especially in NMBB, and the contents of several Bas were much lower than their limits, which also indicated the high quality of natto fermented by the two bacteria.

#### 3.7.2. Changes in Total Biogenic Amines (TBA) Content

As shown in Figure 5C, the TBA content in NMBB was 255.88 mg/kg, which was lower than that in NMB by 17.53 mg/kg. A significant difference (*p* < 0.05) was observed in the changes of TBA contents in 0–30 h. Shukla et al. [61] also found that mixed strains for the fermentation of soybeans reduced the formation of biogenic amines. TBA contents in NMBB were lower than those in NMB. The acid produced by BZ25 inhibited protease activity and reduced the amount of FAAs, the precursor substance for amine synthesis. Given that the acid produced by BZ25 can neutralize amines, the TBA content of NMBB was lower than that of NMB.

#### 3.7.3. Correlation Analysis

As shown in Figure 6, red indicates a positive correlation, while blue indicates the opposite. A positive correlation indicates promoting effect, and a negative correlation indicates inhibiting effect. TBA was significantly positively correlated with TVC of NMB and GUTU09VC of NMBB, with correlation coefficients of 0.878 and 0.808. Bas in natto are mainly metabolized by *Bacillus subtilis* [30]. Figure 6A shows that NK was directly produced by GUTU09. TVC of GUTU09 in NMB promotes the content of TBA, spermine, and tyramine but decreases cadaverine and spermidine. Spermine and tyramine contribute a relatively high proportion of TBA, but cadaverine reduces TBA, which was its metabolic transformation by GUTU09. Figure 6B showed that although BZ25 does not produce NK, it also promoted GUTU09 to produce more NK. BZ25 and GUTU09 promoted each other and jointly resisted changes in unfavorable environments. The VC of GUTU09 in NMBB was positively correlated with tyramine and spermine, negatively correlated with cadaverine and spermidine, and not correlated with putrescine. Comparing Figure 6A,B, it can be found that the addition of BZ25 changes the correlation between indicators. For example, when GUTU09 was fermented alone, TBA, spermine, and NK were positively correlated with pH, but the double-bacteria fermentation of GUTU09 and BZ25 was just right on the contrary. PH drops due to the acid-producing action of BZ25. PH influences some important factors affecting BA formation, namely, enzyme activity and bacterial growth [14]. Both Bas of NMBB and NMB were mainly produced by the metabolism of GUTU09, and this correlation was reduced by the addition of BZ25.

## 4. Conclusions

This study investigated the changes in flavor, the contents of functional compounds, and the biogenic amines of natto, which revealed that the addition of BZ25 improved the taste and flavor of natto, increased the NK and TA content, and decreased the content of BAs. BZ25 was added to GUTU09 to co-fermented soybean, which played a synergistic effect. Correlation analysis showed that the production of pH, TA, Bas, and NK was closely related to the VC, and the addition of BZ25 reduced the correlation between GUTU09 and TBA in natto. The organic acids and NK in NMBB were produced more than those in NMB, and BAs were reduced. Co-fermentation produces natto with a more harmonious aroma than single-strain fermentation. However, additionally, the TAVs of bitter amino acids in the NMBB were decreased as well as total amine content but increased NK activity. All the above results indicated that the quality of natto fermented by mixed bacterial strains was better than that of natto fermented by a single strain. This work laid the foundation for developing new starter cultures and fermentation methods to produce natto products with improved sensory quality and high contents of physiologically active substances, such as NK, which is beneficial to the human body.

## Figures and Tables

**Figure 1 foods-11-02674-f001:**
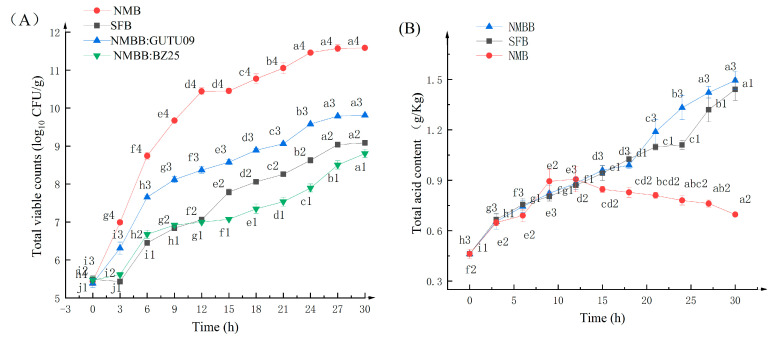
(**A**) Changes in the total viable count of SFB, NMB, and NMBB samples from 0 to 30 h. NMB: viable count of GUTU09 in NMB; SFB: viable count of BZ25 in SFB; NMBB: GUTU09: viable count of GUTU09 in NMBB; NMBB: BZ25: viable count of BZ25 in NMBB. Means with different characters (ai–hi, i = 1, 2, 3 or 4) on the same curve are significantly different (*p* < 0.05), the same below. (**B**) Changes in titratable acid of SFB, NMB, and NMBB samples from 0 to 30 h. (**C**) Changes in the pH of SFB, NMB, and NMBB samples from 0 to 30 h.

**Figure 2 foods-11-02674-f002:**
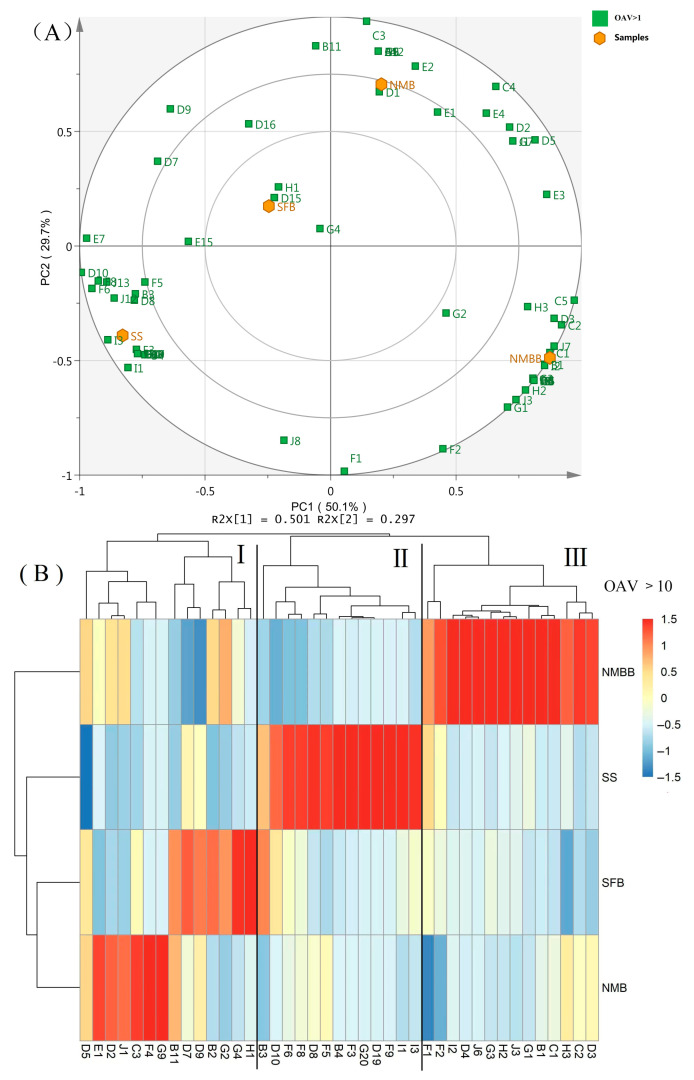
Biplot of principal component analysis in four samples (OAV > 1) (**A**). Gallery plot (**B**), clustering results of steamed soybean (SS), soybean fermented by BZ25 (SFB), natto made by GUTU09 (NMB), and natto made by two mixed strains (NMBB) (OAV > 10). Each rectangle represents a volatile compound. Color represents the OAV of the compound, with blue and red indicating low and high, respectively. The higher the OAV of VOCs the darker color. A refers to amines; B refers to esters; C refers to pyrazines; D refers to ketones; E refers to acids; F refers to aldehydes; G refers to alcohols; H refers to aromatics; I refers to furans and J refers to others volatile compounds.

**Figure 3 foods-11-02674-f003:**
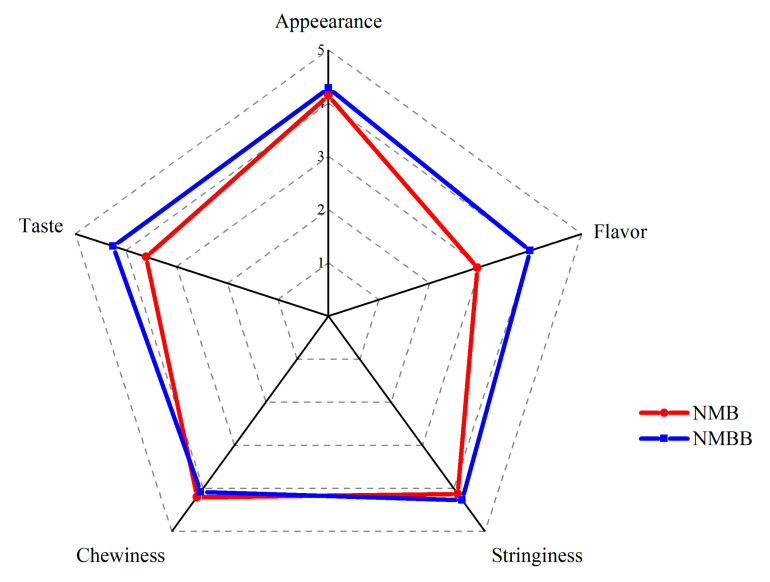
Net chart of sensory properties in natto fermented by two-strain (NMBB) and single-strain fermentation with GUTU09 (NMB).

**Figure 4 foods-11-02674-f004:**
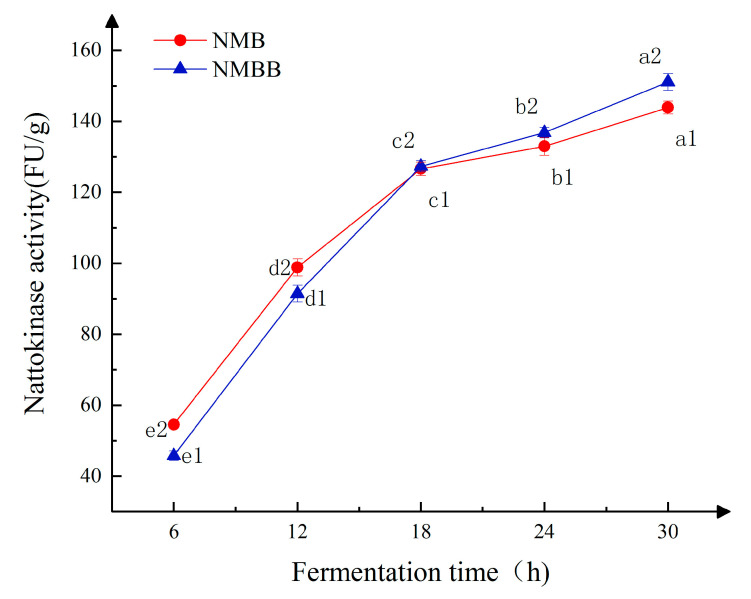
Changes in NK activity during single-strain fermentation and mixed-strain fermentation. Means with dissimilar characters (ai–ei, i = 1 or 2) in the same curve indicate significant differences (*p* < 0.05).

**Figure 5 foods-11-02674-f005:**
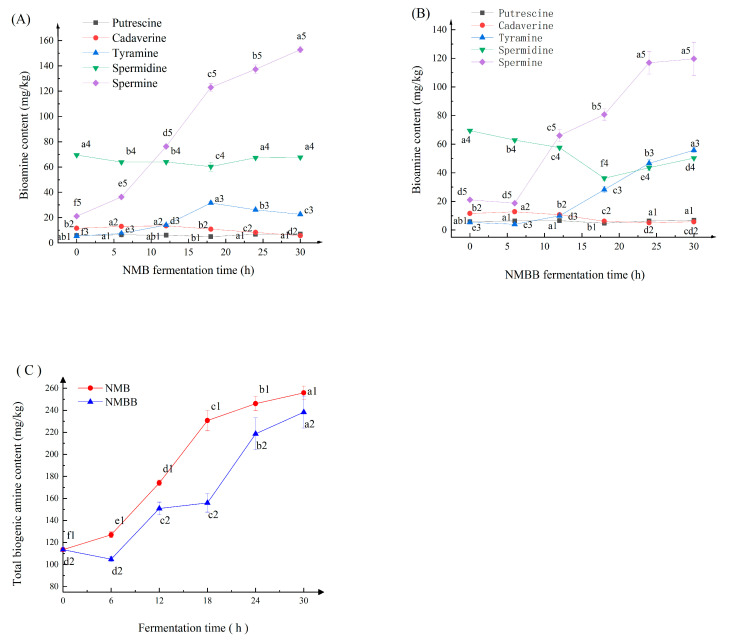
Changes in Bas during natto fermentation by single bacteria (NMB) (**A**) and by two-strain (NMBB) (**B**). Changes in total biogenic amines (TBA) in soybean fermented by single and double bacteria (**C**). Means with different characters (ai–fi, i = 1, 2 ,3, 4 or 5) on the same curve are significantly different (*p* < 0.05).

**Figure 6 foods-11-02674-f006:**
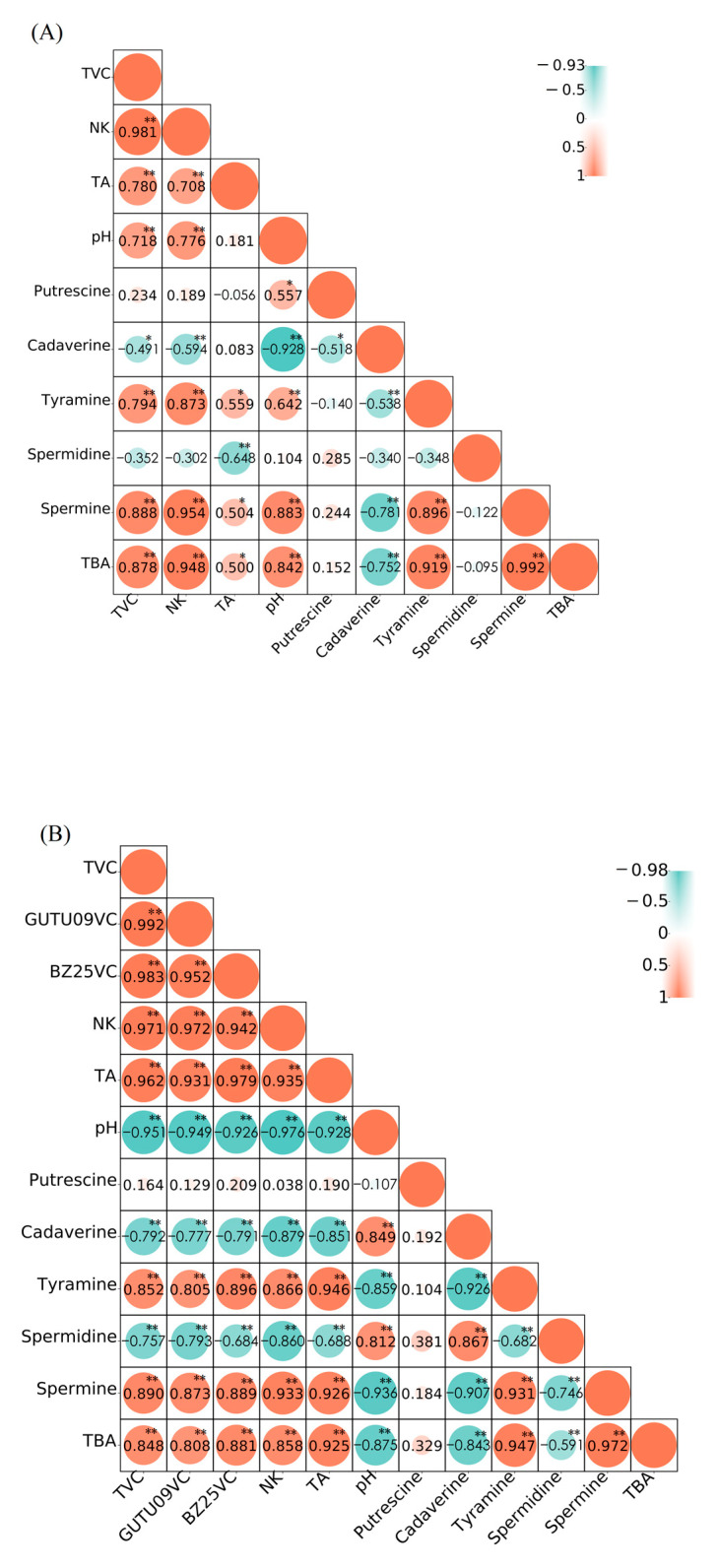
Pearson correlation coefficients between different indexes in natto. NMB (**A**); NMBB (**B**); GUTU09VC refers to the viable count of GUTU09 in the NMBB; BZ25VC refers to the viable count of BZ25 in the NMBB. TVC: total viable count; NK: nattokinase; TA: titratable acidity; “*” *p* < 0.05 was significantly different; “**” *p* < 0.01 was extremely significantly different.

**Table 1 foods-11-02674-t001:** OAV of flavor compounds identified in steamed soybean (SS), soybean fermented by BZ25 (SFB), natto made by GUTU09 (NMB), and natto fermented by two-strain (NMBB).

Number	Name	Aroma Description	RI	ID	Threshold Value (µg/L) ^a^	OAV			
						SS	SFB	NMB	NMBB
	Amines								
A1	*N*,*N*-dimethyl-Methylamine	Fish	2081.3	MS, RI, Std	8–23 ^d^	—	—	7–20	—
	Esters								
B1	Hexanoic acid ethyl ester	Fruit, pineapple, banana aroma	1233.3	MS, RI	55.3 ^b^	<1	<1	10	98
B2	Octanoic acid ethyl ester	Cream, milk	1436	MS, RI	13 ^c^	1	28	4	20
B3	Benzoic acid ethyl ester	Camomile, flower, celery, fruit	1676.6	MS, RI	0.6 ^d^	193	239	—	16
B4	2-Methyl-propanoic acid ethyl ester	Fruity	969	MS, RI	58 ^e^	49	—	3	3
B11	Benzeneacetic acid methyl ester	Jasmine flower, sweet, fruit	1767.6	MS, RI	0.16 ^h^	—	1347	1171	—
	Pyrazines								
C1	Trimethyl-pyrazine	Nutty, cocoa-like, roasted	1407.7	MS, RI	50–96 ^g^	40–77	36–90	62–119	171–328
C2	2,5-Dimethyl-pyrazine	Cocoa, roasted nut, roast beef	1325.7	MS, RI	1700 ^i^	1	1	5	14
C3	2-Ethyl-3,5-dimethyl-pyrazine	Roasted potato	1465	MS, RI	7.5 ^b^	2	74	182	10
C4	2-Ethyl-5-methyl-pyrazine	Nutty	1394.4	MS, RI	16 ^j^	—	8	10	7
	Ketones								
D2	5-Methyl-2-hexanone	—	1141.8	MS, RI	62–89 ^d^	1	2–3	17–24	11–16
D3	2,3-Hexadecanone	Butter, caramel, fruit	2079.1	MS, RI	7.3 ^d^	12	4	33	72
D4	2,3-Butanedione	Sweet, cream	984.7	MS, RI	5 ^e^	—	33	—	644
D5	2-Nonanone	Sweet, coconut	1342.7	MS, RI	32 ^d^	4	42	46	46
D7	2-Heptanone	Mild medicinal fragrance	1181.3	MS, RI	1 ^i^	1287	2376	982	—
D8	3-Octanone	Herb, butter	1251.9	MS, RI	1.3 ^g^	602	24	172	—
D9	2,3-Pentanedione	Caramel, nuts	1061.6	MS, RI	30 ^d^	6	11	7	—
D10	Acetone	irritating flavors	819.8	MS, RI	1100 ^d^	25	16	7	—
D19	6-Methyl-5-hepten-2-one	Pepper, mushroom	1338.9	MS, RI	68 ^d^	16	—	—	—
	Acids								
E1	Acetic acid	Stimulative sour	1451.1	MS, RI	13–150 ^g^	13–485	—	62–715	26–301
E2	2-Ethyl-butanoic acid	Fruit milchigs	1380.5	MS, RI	93–460 ^d^	—	—	6–29	1–7
	Aldehydes								
F1	Decanal	Sweet flowery, wax	1502	MS, RI	0.1 ^i^	5510	3739	—	6675
F2	Nonanal	Flowery, citrus, fat, wax fragrant	1396.1	MS, RI	3.1 ^g^	151	124	40	278
F3	3-Methyl-butanal	Malt, unpleasant smell	919.7	MS, RI	31.6 ^d^	505	—	16	—
F4	2-Methyl-butanal	Cocoa, almond	916.3	MS, RI	1 ^d^	—	—	540	—
F5	Octanal	Crude oil smell	1284.6	MS, RI	0.9 ^j^	316	—	120	—
F6	Phenylacetaldehyde	Fengxinzi taste	1647	MS, RI	4 ^j^	108	41	30	—
F8	Benzaldehyde	Cherry, nuts, bitter almond aroma	1528.8	MS, RI, Std	85 ^g^	80	25	26	—
F9	Hexanal	Grass, fat	1082.9	MS, RI	230 ^g^	22	<1	<1	—
	Alcohols								
G1	1-Octen-3-ol	Mushroom, green	1447.3	MS, RI	10 ^g^	168	13	6	931
G2	Trans-geraniol	Flowery, lemon aroma	1845.3	MS, RI	1 ^f^	—	208	—	189
G3	3-Octanol	Herbaceous, melon, citrus-like odor	1389.8	MS, RI	18 ^d^	—	7	—	39
G4	1-Hexanol	Green, fowery	1350.3	MS, RI	34 ^g^	—	52	—	12
G9	2-Methyl-3-hexanol	—	1370.8	MS, RI	46–81 ^d^	—	—	11–29	—
G20	Eucalyptol	—	1206.7	MS, RI, Std	4.6 ^d^	73	—	—	—
	Aromatics								
H1	2-Methoxy-phenol	Spicy, medicine fragrance	1869.1	MS, RI	1.5 ^g^	465	2200	569	495
H2	Biphenyl	—	2005	MS, RI	0.5 ^d^	9	11	5	65
H3	Phenol	Phenol	2010.6	MS, RI	21 ^g^	25	19	30	37
	Furans								
I1	2-Pentyl-furan	Bean, green	1230.1	MS, RI	5.8 ^i^	502	126	35	78
I2	2,3-Dihydro-benzofuran	—	2399	MS, RI	48 ^d^	2	3	3	13
I3	2-Ethyl-furan	Malt fragrance	957.3	MS, RI	2.3 ^i^	7694	2029	334	—
	Others								
J1	Hexadecane	Alkane	1405	MS, RI, Std	25 ^i^	—	—	84	56
J2	Undecane	Alkane	1090.5	MS, RI, Std	1578 ^i^	—	—	<1	<1
J3	Styrene	Balsamic, gasoline	1259.6	MS, RI	65 ^d^	2	2	—	16
J6	Naphthalene	—	1757.7	MS, RI	6 ^d^	—	—	—	43
J7	Anethole	Fennel, spicy, licorice smell	1839	MS, RI	100 ^f^	—	—	4	20
J13	Dimethyl trisulfide	Sulfuric, strong onion odor	1380.1	MS, RI	14 ^g^	19	19	2	—

Note: ^a^ Odor thresholds were determined in water. Odor thresholds were taken from reference. ^b^ Odor thresholds taken from Ref. [42]; ^c^ Odor thresholds taken from Ref. [43]; ^d^ Odor thresholds taken from Ref. [44]; ^e^ Odor thresholds taken from Ref. [45]; ^f^ Odor thresholds taken from Ref. [46]; ^g^ Odor thresholds taken from Ref. [7]; ^h^ Odor thresholds taken from Ref. [47]; ^i^ Odor thresholds taken from Ref. [48]; ^j^ Odor thresholds taken from Ref. [49]. OAV: odor activity value; SS: steamed soybean; SFB: soybean fermented by BZ25; NMB: natto made by GUTU09; NMBB: natto made by two mixed strains; ID: Identification basis; MS: mass spectrometry; RI: retention index; Std: authentic standards; “n.f.”, “—”: Data was not found in literatures.

## Data Availability

Data is contained within the article and Supplementary Materials.

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
