# Peer review of "Effect of Adding Bifidobacterium animalis BZ25 on the Flavor, Functional Components and Biogenic Amines of Natto by Bacillus subtilis GUTU09"

_foods, 2022, doi:10.3390/foods11172674_

Round 1

Reviewer 1 Report

The paper was focused on the determination and comparison of volatile organic compounds (VOCs), free amino acids (FAAs) and biogenic amines (BAs), nattokinase (NK) of natto made by fermenting soybean with a single strain of Bacillus subtilis, and respectively, two-strains of Bifidobacterium animalis and Bacillus subtilis.

Despite a large number of analytical determinations, the article has a poor discussion of the results. The authors might take into consideration rewriting the Results and Discussion section in a more critical manner.

Some aspects that need to be corrected are:

The Introduction section

-a paragraph discussing the usual way of natto fermentation would be necessary; some information regarding the type of microorganisms involved in fermentation, at least

-page 2, line 4-please correct “erecipe

-page 2, lines 79-80 “The present work firstly utilized mixed fermented as fermenters to improve the flavor and taste of natto” it seems that this sentence is incomplete, especially when it has to describe the paper’s aim.

The entire paragraph (lines 77-83) needs to be rephrased in order to better introduce the purpose of the work

The Materials/Methods section

-page 3, lines 99-101- please add information regarding the microbial strains used in the study (starter culture or not; producer, lyophilized?  etc)

- page 3, lines 112-114- please use italic for microorganisms

-I suggest introducing the samples’ codification in the Materials /Methods section because is easier for the readers to go through the manuscript and for a better understanding

-page 3- line 113,115-please define the abbreviation LB; this observation is valid for all abbreviations at the first mention of its

-page 3, lines 119-121- are you sure that TTA expression as lactic acid is correct? Usually, TTA is mL of NaOH used for titration. If it would be expressed as lactic acid, a conversion coefficient will be necessary!

-page 4, line 155- even if the sensory analysis was made according to another study, information regarding the panelists used in this work is mandatory (number of panelists, trained or untrained, age, gender etc) and the authors must mention if the sensorial study was approved.

Results and Discussion section

-page 9, line 259- Steamed soybean (SS) is firstly mentioned in the results! The authors need to present and explain clearly the entire research design in the Materials and methods section

-page 9, line 273-please correct “tosynthesis

The results’ discussion is poor; very few citations from other research papers are used for the discussion. For example, the microbial count, TTA, and pH are presented without any comparisons with other results

-page 9- subsections’ numbering use 3.2 twice

Author Response

A detailed list (point to point) of responses to each item of the review comments

To Reviewer 1

Q: a paragraph discussing the usual way of natto fermentation would be necessary; some information regarding the type of microorganisms involved in fermentation, at least.

A: Thanks. I added the usual way of natto that would be necessary, Some information regarding the type of microorganisms involved in fermentation this part of content.

The section (lines 31-33 - in original manuscript) “Natto is a traditional fermented soybean product that contains various functional components, such as nattokinase (NK), vitamin K, isoflavones, superoxide dismutase, phospholipids, saponins, and linoleic acid [1,2].” has been modified as follows: Natto is a traditional fermented soybean product fermented by Bacillus subtilis that contains various functional components, such as nattokinase (NK), vitamin K, isoflavones, superoxide dismutase, phospholipids, saponins, and linoleic acid [1,2]. Natto is a low-cost but high-nutrition product made using a simple procedure. Presoaked soybeans are cooked until tender, drained, cooled to 40 °C, inoculated with Bacillus natto, and incubated at 30 – 40 °C for 12 - 36 h. The production process of fermented soybeans mainly includes selecting soybeans, soaking, sterilization, steaming, inoculation fermentation and after-ripening [3,4]. (Lines 31-38)

Q: page 2, line 4-please correct “erecipe”

A:Thanks. We have modified it. The word “errecipe” has been replaced by “recipe” (Lines 53).

Q: page 2, lines 79-80 “The present work firstly utilized mixed fermented as fermenters to improve the flavor and taste of natto” it seems that this sentence is incomplete, especially when it has to describe the paper’s aim. The entire paragraph (lines 77-83) needs to be rephrased in order to better introduce the purpose of the work.

A: Thanks. We have substantially modified the section. The section (page 2, lines 79-80 - in original manuscript) “The present work firstly utilized mixed fermented as fermenters to improve the flavor and taste of natto.” has been modified as follows: The present work firstly utilized mixed starters of Bacillus subtilis GUTU09 and Bifidobacterium animalis BZ25 to improve the flavor and taste of natto (Lines 89).

The section (page 2, lines 77-83 - in original manuscript) “Therefore, there are still many defects in the research on soybean fermented by GUTU09. Much research on natto was limited to the improvement of function, without considering the issue of consumer acceptance at the same time. The present work firstly utilized mixed fermented as fermenters to improve the flavor and taste of natto. Secondly, the process of NK and BAs changes during fermentation was investigated to explore the feasibility of adding probiotic BZ25 to improve the flavor and functional components of natto.” has been modified as follows: The present work firstly utilized mixed starters of Bacillus subtilis GUTU09 and Bifidobacterium animalis BZ25 to improve the flavor and taste of natto. The volatile substances in natto were analyzed by coupling headspace solid-phase microextraction (HS-SPME) and gas chromatography-mass spectrometry (GC-MS). HPLC was utilized to detect free amino acids to investigate the change of bitter amino acid content in natto. Secondly, NK and BAs changes during fermentation were investigated to explore the feasibility of adding probiotic BZ25 to improve the flavor and functional components of natto (Lines 89 - 96).

Q: page 3, lines 99-101- please add information regarding the microbial strains used in the study (starter culture or not; producer, lyophilized?  etc)

A: Thanks. I have added basic information about the strains to the original text. We used seed culture. The section Bacillus subtilis GUTU09 was isolated from the traditional fermented soybeans, in Guizhou Province, China. BZ25 was from Guizhou Xiang Pigs.” was added (Lines 121-123).

Q: page 3, lines 112-114- please use italic for microorganisms

A: Thanks. We have done so (Lines 139-140).

Q: I suggest introducing the samples’ codification in the Materials /Methods section because is easier for the readers to go through the manuscript and for a better understanding.

A: Thanks. I've introduced the samples’ codification in the Materials / Methods. The section “In this work, steamed soybean (SS) samples were obtained after sterilization of soaked soybeans. Steamed soybean (SS), soybean fermented by BZ25 (SFB) and natto made by GUTU09 (NMB) were used as the controls, and natto made by two mixed strains samples (NMBB) were made and analyzed.” was added (Lines 131 - 134).

Q: page 3- line 113,115-please define the abbreviation LB; this observation is valid for all abbreviations at the first mention of its.

A: Thanks, We have modified it. The word “LB” have replaced by “Luria-Bertani (LB)” (Lines 137).

Q: page 3, lines 119-121- are you sure that TTA expression as lactic acid is correct? Usually, TTA is mL of NaOH used for titration. If it would be expressed as lactic acid, a conversion coefficient will be necessary!

A: Thanks. Bifidobacterium mainly produce lactic acid [1]. Bujna, et al. [2] found that Bifidobacterium BB-46 metabolized 4.5-7 times more lactic acid than other acids. The original manuscript data results have been multiplied by the lactic acid coefficient (0.09). The description in the method was missing and has been modified. (page 3, lines 119-121- in original manuscript)  The section “Titratable acidity was measured by titration with 0.01 M sodium hydroxide based on the national standards method GB/T 12456-2008, and it was expressed as the content of lactic acid.” has been modified as follows: Titratable acidity was measured by titration with 0.01 M sodium hydroxide based on the national standards method GB/T 12456-2021. It was expressed as the content of lactic acid, and the conversion factor is 0.09. (lines 146-147).

Q: page 4, line 155- even if the sensory analysis was made according to another study, information regarding the panelists used in this work is mandatory (number of panelists, trained or untrained, age, gender etc) and the authors must mention if the sensorial study was approved.

A: Thanks. We have made some modifications. The section (page 4, line 155 - in original manuscript) “The sensory characteristics of natto was evaluated using the procedure by Yang, et al. [26].” has been modified as follows: The sensory characteristic of natto were evaluated using the procedure by Yang, et al. [3]. A sensory team consisted of 20 teachers and graduate students majoring in food-related courses with sensory evaluation experience. The samples were labeled with different numbers, and then randomly disrupted. The sensory characteristics of natto were mainly determined by its appearance, stickiness, flavor, taste, and chewiness. The appearance of natto was evaluated by darkness and uniformity. The stickiness was measured by how well natto clung to chopsticks. The flavor of natto was evaluated by olfaction, ammonia odor, and bean odor. Good natto should have a plain or slightly sour smell and taste, whereas an ammonium odour and a dead leaf odour are undesirable. The taste was expected to be slightly acidic or mellow. The chewiness is the feedback of teeth on the softness, hardness, stickiness, and smoothness of natto. For all the sensory traits, rating scores from 1 to 5 were used, where 5 = excellent, 4 = good, 3 = moderate, 2 = poor, and 1 = inferior. Finally, a high index score indicated good natto quality (lines 185-196).

 Q: page 9, line 259- Steamed soybean (SS) is firstly mentioned in the results! The authors need to present and explain clearly the entire research design in the Materials and methods section.

A: Thanks. We have added as follows: In this work, steamed soybean (SS) samples were obtained after sterilization of soaked soybeans (lines 131-132).

Q: page 9, line 273-please correct “tosynthesis”.

A: Thanks. The word “tosynthesis” has been replaced by “be synthesized” (line 348).

Q: The results’ discussion is poor; very few citations from other research papers are used for the discussion. For example, the microbial count, TTA, and pH are presented without any comparisons with other results.

 A: Thanks. We have substantially modified the section. We quite agree with the reviewer's comments, The section (lines 222-448 - in original manuscript) “Given that strain types and production processes can affect fermentation and then change the composition of aroma components.…the acid produced by BZ25 inhibited protease activity and reduced the amount of FAAs, the precursor substance for amine synthesis. Given that the acid produced by BZ25 can neutralize amines, TBA content of NMBB was lower than NMB.” has been modified as follows: 3.3 Effect of BZ25 addition on the volatile compounds of natto

Given that strain types and production processes can affect fermentation and then change the composition of aroma components. SS, SFB, NMB, and NMBB were identified by HS-SPME-GC-MS. As shown in Table S1, 133 kinds of volatile compounds contents were identified. It was found that the increase of TA content and the decrease of pH in the sample did not increase the content of volatile acids. The reason was maybe that the increased acid is mainly non-volatile acid. To further examine the contribution of these 133 volatile compounds to the overall aroma profile of natto, their OAVs were calculated according to their concentrations and odor thresholds (Table S2).

As shown in Table 1, volatile substances with OAV > 10 in Table S2 are listed separately in order to more clearly find the substances that contribute significantly to the overall flavor of the sample. Aldehydes are mainly derived from lipid oxidation and Strecker amino acid degradation [4]. Precisely, linear fatty aldehydes and alkene aldehydes usually originate from lipid oxidation. Decanal, nonanal, hexanal and octanal originate from the oxidation of unsaturated fatty acids. In contrast, branched-chain aldehydes, such as benzaldehyde, 3-methylbutyraldehyde, and phenylacetaldehyde, are generated by the degradation of Strecker amino acids [5]. Due to the low flavor threshold, aldehydes contribute significantly to the unique flavor of natto products. Ketones can be generated by amino acids or by the Maillard reaction [6]. Alcohols may be produced from the metabolism of unsaturated fatty acids of microorganism [7]. Pyrazines can be generated by the Maillard reaction or the thermal reaction of microbial metabolites [8]. Acids are the lipid hydrolysate of soybeans, and have a sour or oily odor [9]. Organic acid volatiles may also be another cause of the pungent smell of natto. In the presented study, most of amines have a stimulating taste, which was the main reason for the ammonia flavor of natto. N,N-dimethyl-methylamine, formamide, 3-methyl-butanamide, ethosuximide were detected in NMB, but not in NMBB. Natto fermented by GUTU09 produces an ammonia smell, which is consistent with the report from Chen, et al. [10]. The disappearance of amines indicated that ammonia taste of natto was effectively reduced. The volatile substances of NMBB mainly included alcohols (G) and ketones (D) aldehydes (F). Trimethyl-pyrazine (nutty, cocoa, and roasted flavors), 2,3-butanedione (sweet cream), decanal (sweet and flowery), 1-octen-3-ol (mushroom) were the main volatile substances in NMBB. NMB mainly included esters (B) and ketones (D) aldehydes (F). Liu, Su and Song [6] found that most esters were described as having fruit and floral flavors, such as pineapple, nut, banana, and apple flavors. It is speculated that acids and alcohols can be synthesized a large number of esters via esterification [11]. Bifidobacterium was added to fermented soybean, which resulted in significant changes in the important characteristics and flavor of natto. The reduction of volatile acids was consistent with the changes of FAA content, which may be converted into other organic compounds. After the addition of BZ25, volatile amine disappeared, volatile aldehyde and ketone species decreased, and alcohols increased. Evidently, the volatile components of natto were usually related to microbial metabolism [12]. Compared with the sensory evaluation results, it is the disappearance of some irritating volatile substances that makes the NMBB sample more popular. Volatile substances of NMBB had great changes in esters, pyrazines, ketones, acids, aldehydes, alcohols, amines, and aromatics, and according to sensory scores, these changes were developed in a positive direction. These changes indicated that co-fermentation could effectively improve the flavor of natto. He, et al. [12] found that the co-fermentation of three strains during natto fermentation may facilitate the formation of flavors.

 3.3.1 Principal component analysis (PCA) and Hierarchical cluster analysis (HCA)

To visualize a total picture of the distributions of 73 odor-active compounds (OAV > 1) in all samples, PCA was applied (Fig. 2A). PCA showed the major differences among the volatile profiles of four samples. The first principal component (PC1) of the PCA accounted for 50.1 %, and PC2 explained an additional 29.7 % of the total variability. And the cumulative contribution rate reached 79.8%. PCA Score Plot (Fig. 2A) shows that NMB and SFB were located in the first and second quadrants and positively correlates with PC2. SS and NMBB were located in the third and fourth quadrants and negatively correlate with PC2. Four samples were distributed in different quadrants, indicating that there were significant differences in flavor substances between samples. The characteristic components of SS were 2-methyl-propanoic acid ethyl ester (B4), 3-octanone (D8), 3-methyl-butanal (F3), octanal  (F5), phenylacetaldehyde (F6), eucalyptol (G20), 2-pentyl-furan(I1), 2-ethyl-furan (I3), those of SFB were 1-hexanol (G4) and 2-methoxy-phenol(H1), thoseof NMB was N,N-dimethyl-Methylamine (A1),  2-Ethyl-3,5-dimethyl-pyrazine (C3), acetic acid (F1),  2-ethyl-butanoic acid (F2) and 2-methyl-butanal (G4), and those of NMBB were hexanoic acid ethyl ester (B1), trimethyl-pyrazine (C1) 2-hexadecanone (D3), 2,3-butanedione (D4), nonanal (F2), 3-octanol (G3), biphenyl (H2) and naphthalene (J6). Therefore, the characteristic flavor components of natto can be changed by mixed fermentation of two bacteria.

Fig. 2(B) shows that hierarchical clustering and heat map visualization of volatile compounds of four samples in different treatments (OAV > 10). The gallery plot of four samples was applied to visualize the differences of aroma compounds. The aroma components of natto produced by co-fermentation were different from those produced by single strain GUTU09. Next, we used clustering analysis for Volatile compounds content and found that the four groups could be divided into three categories. Class I (SFB and NMB) included high content of acetic acid (E1), 2-ethyl-3,5-dimethyl-pyrazine (C3), 2-methyl-butanal (F4), 1-hexanol (G4), 2-methyl-3-hexanol (G9) and 2-methoxy-phenol (H1). Class II (SS) contained high content of 2-pentyl-furan (I1), 2-ethyl-furan (I3), hexanal (F9), 6-methyl-5-hepten-2-one (D19), eucalyptol (G20), 3-methyl-butanal (F3), 2-mmethyl-propanoic acid ethyl ester (B4), octanal (F5), 3-octanone (D8) and phenylacetaldehyde (F6). Class III (NMBB) mainly included high content of 2,3-hexadecanone (D3), 2,5-dimethyl-pyrazine (C2), trimethyl-pyrazine (C1), hexanoic acid ethyl ester (B1), 1-octen-3-ol (G1), styrene (J3), biphenyl (H2), 3-octanol (G3), naphthalene (J6), 2,3-butanedione (D4) and 2,3-dihydro-benzofuran (I2). Fig. 2(B) shows that the volatile substances of NMBB were accumulated more than NMB, and the flavor substances that contribute more to the whole are very different from NMB. PCA plot and gallery map show that mixed fermentation can greatly change the composition of volatile compounds of natto, thus affecting the sensory score and increasing people's acceptance of natto.

3.4 Sensory properties

The sensory properties of NMBB and NMB were evaluated in terms of appearance, stringiness, flavor, taste, and chewiness. As shown in Fig. 3, there was no significant difference in appearance, stringiness and chewiness scores between the two samples. The flavor of NMBB was 3.98, while that of NMB was 2.94. The taste score of the NMBB was 4.25, while that of NMB was 3.6. NMBB was presented with sour, and roasted nutty fragrance. NMB showed ammonia, fishy, fruity, cocoa fragrance. During fermentation, the amounts of ammonia are released to give a negative influence on the flavor of natto [10]. The disappearance of amine volatiles and the decrease of bitter amino acids make NMBB more popular. The increase in flavor score was attributed to the lactic acid produced by BZ25, which neutralized amines and gave natto a slightly acidic odor. BZ25 plays a major role in reducing the ammonia taste and bitter taste of NMBB. The sensory properties of fermented soybean products are affected by various microorganisms and microbial enzymes also influenced the characteristic taste and flavor of products [13]. Soy protein can be hydrolyzed by proteases to produce peptides and amino acids. The bitter taste of natto during chewing may be related to bitter peptides and amino acids such as leucine, histidine, and proline, especially peptides and tyrosines produced by casein hydrolysis [14]. From the data of FAAs (Table 2), it can be seen that the mixed fermentation of two bacteria can reduce the bitter taste of natto. The decrease of bitter amino acid content is also one of the reasons for the increase of taste of NMBB. The sensory evaluation results showed that the mixed fermentation was beneficial to improve the flavor of natto.

3.5 Effect of BZ25 addition on FAAs and bitter amino acids

The FAA contents of four samples were detected and shown in Table 2. FAA content affects the quality of food products [15]. Bitter amino acids are rated as the leading influencing factors of poor taste [16]. FAAs in food, such as lysine, threonine and methionine, affect the protein quality and flavor of food [17]. Eight bitter amino acids detected in NMBB and NMB. Except His, the TAV of bitter amino acids (NMBB) was lower than that of NMB. In both groups of samples, Lys had the highest TAV, which was 7.0 and 9.9, respectively. In natto, the acid produced by BZ25 reduced pH. It also decreased the capability of protease to hydrolyze protein into FAAs by inhibiting protease activity. The content of FAAs can reflect the degree of protein degradation in soybean samples [18]. Proteases were produced during the growth and metabolism of GUTU09. These enzymes promoted the hydrolysis of proteins into various peptides, such as short peptides, and amino acids. These substances could be decomposed and oxidized by BZ25 and used as nitrogen sources for bacterial growth, which led to the reduction in FAA content. This effect could be another reason for the reduction in FAA content. The above data showed that two-strain mixed fermentation could decrease the content of bitter amino acids.

3.6 Effect of BZ25 addition on NK activity

As shown in Fig. 4, a significant difference was observed in the changes of NK activity (P<0.05). NK activity gradually increased with the extension of fermentation time. Before 18 h, The NK activity in NMB was higher than that in NMBB sample. The NK activities in both groups of samples were similar at 18 h. After more than 18 h of fermentation, NMBB showed higher NK activity than NMB. From Fig. 1, the number of viable cells of GUTU09 in NMB is much higher than that of NMBB, and it is obvious that NK is produced by GUTU09. This indicates that it is not that the more viable bacteria GUTU09 is, the higher its NK-producing activity, and its NK-producing ability is also regulated by other factors, such as BZ25. Chen, et al. [19] found that the total phenolic and genistein contents were increased by co-fermentation of Lactobacillus with Bacillus subtilis. In the later stage of fermentation, BZ25 promoted the production of NK by GUTU09. NK increased from 143.89 FU/g to 151.05 FU/g after 30 h of fermentation. In natto, the addition of BZ25 promoted the accumulation of NK and the increment in active substance content and quality.

3.7 Effect of BZ25 addition on BAs

3.7.1 Changes in BAs during fermentation

As shown in Fig. 5(A and B), only five BAs including putrescine, cadaverine, tyramine, spermidine, and spermine, were detected in NMB and NMBB. A significant difference (P<0.05) was observed in the changes of spermine and tyramine contents (Fig.5). Cadaverine and putrescine have no direct toxicity to the human body and are considered natural amines in food [20]. Tyramine contents exceeding 100 mg/kg and β-phenylethylamine contents over 30 mg/kg are considered to be toxic doses [21]. Histamine and 2-phenylethylamine were not detected in all samples. The highest content of tyramine was 55.78 ± 1.79 mg/kg, which did not exceed limits, indicating that the strain used for fermentation had good biological safety.

The change curves of putrescine and cadaverine in NMBB and NMB were relatively flat as fermentation progressed. The spermine content in NMB showed an increasing trend and that in NMBB sample decreased slightly at first and then increased continuously. From 0 h to 30 h, the spermine content in NMBB increased from 21.12 ± 0.22 mg/kg to 119.65 ± 1.66 mg/kg. The minimum concentration of spermine required to exert cytotoxicity was 653.56 mg/kg [22]. The spermine content in NMBB sample was 119.65 mg/kg, which was far below the minimum toxic concentration. Biogenic amines are common in fermented soybean foods. High levels of biogenic amines were found in 21 different natto in Korea [23], among which, 9 natto had β-phenylethylamine or tyramine contents higher than the toxic dose (30 mg/kg and 100 mg/kg, respectively) of each amine. However, the content of each biogenic amine in NMB and NMBB did not exceed its limit, especially that in NMBB, and the contents of several biogenic amines was much lower than their limits, which also indicated the high quality of natto fermented by the two bacteria.

3.7.2 Changes in total biogenic amines (TBA) content

As shown in Fig. 5C, the TBA content in NMBB was 255.88 mg/kg, which was lower than that in NMB by 17.53 mg/kg. A significant difference (P<0.05) was observed in the changes of TBA contents in 0 - 30 h. Shukla, et al. [24] also found mix strains for the fermentation of soybeans reduced the formation of biogenic amines. TBA contents in NMBB were lower than those in NMB. The acid produced by BZ25 inhibited protease activity and reduced the amount of FAAs, the precursor substance for amine synthesis. Given that the acid produced by BZ25 can neutralize amines, TBA content of NMBB was lower than that of NMB (lines 279-613).

Q: page 9- subsections’ numbering use 3.2 twice

A: Thanks. I have corrected it. (line 316,495)

References

  1. Buruleanu, C.L.; Nicolescu, C.L.; Avram, D.; Manea, I.; Bratu, M.G. Effects of yeast extract and different amino acids on the dynamics of some components in cabbage juice during fermentation with Bifidobacterium lactis BB-12. 2012, 21, 691-699. https://10.1007/s10068-012-0090-5.
  2. Bujna, E.; Farkas, N.A.; Tran, A.M.; Dam, M.S.; Nguyen, Q.D. Lactic acid fermentation of apricot juice by mono- and mixed cultures of probiotic Lactobacillus and Bifidobacterium strains. 2018, 27, 547-554. https://10.1007/s10068-017-0269-x.

Reviewer 2 Report

The manuscript intitled “Effect of adding Bifidobacterium animalis BZ25 on the flavor, functional components and biogenic amines of natto by Bacillus subtilis GUTU09” investigates mainly the sensory properties of a fermented product (natto) by a microbial consortium in comparison to the traditional use of one bacterial species. There are important results reported, with the correct use of experimental procedures and conclusions are supported by data. However, some points should be addressed, and an English revision must be performed.

The main innovation of this work is the evaluation of the sensory attributes of the fermented product when performed by a consortium of two bacterial strains. In this sense, my first critical point to the work would be the complete description of the sensory analysis that is missing in the paper.

 Format and text

- The text should be revised to reduce mistakes like: “Natto is a high-value fermented soybean PRODUCED by B. subtilis.” (First line of abstract). This is just an example. An English revision should also be performed.

 Line 46: typing mistake

 - Some abbreviations are not specified in the text. Examples: MRS, LB, GC-MS, SFB…Mainly: in Figure and Table legends all abbreviations should be again specified because they must be understood without text information.

 - Figure 1 and 2 should be better organized to be in one page. In fact there is a blank page and the legend is two pages ahead of the first figure.

 - Table footnote should be revised. All abbreviations should be defined there.

 - Space between lines changes during the document.

 - Figures 1, 4 and 5A and 5B should be improved with the quality of Figure 5C.

 Scientific points

 Introduction section

- “Currently, there are only a few reports on natto fermentation by mixed strains.” References shloud be cited here.

 Methodology

- Section 2.2: Why different inoculum volume was used for the mixed culture (7%) experiment and single-strain culture (3.5%). I believe that for a fair comparison it should be the same.

- Section 2.5: Methodology should be described in detail since it is an important part of the work. Specially parameters that are different from the paper used as reference: number of panelists, how samples were given to them, etc.

- Section 2.7: NK activity should also be detailed here.

 Results

- Table 1 is too long and difficult to “read”. I believe that if you separate each chemical function in one table and only present the important ones (that are discussed in the text) it would improve the clarity of the Table.

 - Section 3.2 should be better discussed. Comparison to sensory results of fermented products in literature.

Author Response

A detailed list (point to point) of responses to each item of the review comments

To Reviewer 2

Q: The main innovation of this work is the evaluation of the sensory attributes of the fermented product when performed by a consortium of two bacterial strains. In this sense, my first critical point to the work would be the complete description of the sensory analysis that is missing in the paper.

A: Thanks. We have substantially modified the section. (Lines 359-368 - in original manuscript) The section “Sensory properties of two natto samples were evaluated in terms of appearance, stringiness, flavor, taste, and chewiness. As shown in Fig. 3, there was no significant dif-ference in appearance, stringiness and chewiness scores between the two samples. Flavor of the two-strain mixed fermented samples was 3.98, while that of NMB sample was 2.94. Taste score of the NMBB sample was 4.25, while that of NMB sample was 3.6. The main reason was the disappearance of amine volatiles and the decrease of bitter amino acids, which makes the NMBB sample more popular. The sensory evaluation results showed that the mixed fermentation method was beneficial to improve the flavor of natto.” has been modified as follows: The sensory properties of NMBB and NMB were evaluated in terms of appearance, stringiness, flavor, taste, and chewiness. As shown in Fig. 3, there was no significant difference in appearance, stringiness and chewiness scores between the two samples. The flavor of NMBB was 3.98, while that of NMB was 2.94. The taste score of the NMBB was 4.25, while that of NMB was 3.6. NMBB was presented with sour, and roasted nutty fragrance. NMB showed ammonia, fishy, fruity, cocoa fragrance. During fermentation, the amounts of ammonia are released to give a negative influence on the flavor of natto [1]. The disappearance of amine volatiles and the decrease of bitter amino acids make NMBB more popular. The increase in flavor score was attributed to the lactic acid produced by BZ25, which neutralized amines and gave natto a slightly acidic odor. BZ25 plays a major role in reducing the ammonia taste and bitter taste of NMBB. The sensory properties of fermented soybean products are affected by various microorganisms and microbial enzymes also influenced the characteristic taste and flavor of products [2]. Soy protein can be hydrolyzed by proteases to produce peptides and amino acids. The bitter taste of natto during chewing may be related to bitter peptides and amino acids such as leucine, histidine, and proline, especially peptides and tyrosines produced by casein hydrolysis [3]. From the data of FAAs (Table 2), it can be seen that the mixed fermentation of two bacteria can reduce the bitter taste of natto. The decrease of bitter amino acid content is also one of the reasons for the increase of taste of NMBB. The sensory evaluation results showed that the mixed fermentation was beneficial to improve the flavor of natto (lines 497-516).

Q: The text should be revised to reduce mistakes like: “Natto is a high-value fermented soybean PRODUCED by B. subtilis.” (First line of abstract). This is just an example. An English revision should also be performed.

A: Thanks. We have done so. The “produced” word has been added (line 14). We have invited people who are proficient in English to polish the article without changing the meaning of the original text. All modified text has been highlighted.

Q: Line 46: typing mistake

A:Thanks. We have modified it. The word “errecipe” has been replaced by “recipe” (Line 53).

Q: Some abbreviations are not specified in the text. Examples: MRS, LB, GC-MS, SFB…Mainly: in Figure and Table legends all abbreviations should be again specified because they must be understood without text information.

 A:Thanks. We have modified it. (Lines 116, 137, 93)

Q: Figure 1 and 2 should be better organized to be in one page. In fact there is a blank page and the legend is two pages ahead of the first figure.

 A: Thanks. We have done so. (page 7, 11)

Q: Table footnote should be revised. All abbreviations should be defined there.

 A: Thanks. We have done so. (Lines 448-449, 543-544)

Q: Space between lines changes during the document.

 A:Thanks. I have checked the full text and corrected the line spacing problem.

Q: Figures 1, 4 and 5A and 5B should be improved with the quality of Figure 5C.

 A: Thanks. We have done so. (page 7, 25, 27)

Q: “Currently, there are only a few reports on natto fermentation by mixed strains.” References shloud be cited here.

 A: Thanks. I deleted the sentence and the section “Currently, how Bacillus subtilis single bacteria, multiple strains affect the fermentation characteristics of natto is unclear [14]. Determining such effect is helpful in understanding the role of strains in soybean fermentation for the preparation of high-quality natto.” was added, because I regret my inaccurate expression. (Lines 55-58)

Q: Section 2.2: Why different inoculum volume was used for the mixed culture (7%) experiment and single-strain culture (3.5%). I believe that for a fair comparison it should be the same.

A: Thanks. Seven pecent inoculum amount fo mixed fermentation was 3.5 % GUTU09 and 3.5 % BZ25, in order to ensure the same inoculum amount of GUTU09 both in the mixed culture and single-strain culture.

Q: Section 2.5: Methodology should be described in detail since it is an important part of the work. Specially parameters that are different from the paper used as reference: number of panelists, how samples were given to them, etc.

A: Thanks. The section (section 2.5 - in original manuscript) “The sensory characteristics of natto were evaluated using the procedure by Yang, et al. [26].” has been modified as follows: The sensory characteristic of natto were evaluated using the procedure by Yang, et al. [4]. A sensory team consisted of 20 teachers and graduate students majoring in food-related courses with sensory evaluation experience. The samples were labeled with different numbers, and then randomly disrupted. The sensory characteristics of natto were mainly determined by its appearance, stickiness, flavor, taste, and chewiness. The appearance of natto was evaluated by darkness and uniformity. The stickiness was measured by how well natto clung to chopsticks. The flavor of natto was evaluated by olfaction, ammonia odor, and bean odor. Good natto should have a plain or slightly sour smell and taste, whereas an ammonium odour and a dead leaf odour are undesirable. The taste was expected to be slightly acidic or mellow. The chewiness is the feedback of teeth on the softness, hardness, stickiness, and smoothness of natto. For all the sensory traits, rating scores from 1 to 5 were used, where 5 = excellent, 4 = good, 3 = moderate, 2 = poor, and 1 = inferior. Finally, a high index score indicated good natto quality (lines 185-196).

 Q: Section 2.7: NK activity should also be detailed here.

 A: Thanks. The section (section 2.7- in original manuscript) “NK activity (FU/g) was analyzed through the fibrin degradation method in accordance with our previous report [5].” has been modified as follows: Ten grams of fermented soybeans were dissolved in 90 mL deionized water and homogenized in a beating machine for 10 s, then extracted at 4 °C for 24 h, and centrifuged at 10,000 × g for 10 min. The sample supernatant was collected for analysis of the NK activity (FU/g) of fermented soybeans through fibrin degradation method in accordance with our previous report [5]. The specific steps were as follows: 1.4 mL of Tris-HCl (50 mM, pH 7.8) buffer and 0.4 mL of fibrinogen solution (7.2 mg/mL) were added to test tube in 37 °C water bath for 5 min, followed by adding 0.1 mL of thrombin (20 U/mL) and then bathed in water at 37 °C for 10 min to form artificial thrombosis. Next, adding 0.1 mL of the sample supernatant to the tube, it was continued to bath in water at 37 °C for 60 min, and then the reaction was terminated by adding 2mL of trichloroacetic acid (0.2 moL/L) solution standing for 20 min. Subsequently, the reaction solution was centrifuged at 10,000 × g for 10 min, and the supernatant was analyzed at 275 nm wavelength by a spectrophotometer (UV-2700, Shimadzu Enterprise Management Co., Ltd.). Definition of enzyme activity: the amount of enzyme required for an increase of 0.01 absorbance at 275 nm per minute is defined as one FU of fibrin degradation enzyme activity (Lines 223-238).

Q: Table 1 is too long and difficult to “read”. I believe that if you separate each chemical function in one table and only present the important ones (that are discussed in the text) it would improve the clarity of the Table.

 A: Thanks. We have replaced the original table 1 with Table S2, and put the flavor substances (OAV > 10) in Table 1. (page 11)

Q: Section 3.2 should be better discussed. Comparison to sensory results of fermented products in literature.

A: Thanks. We quite agree with the reviewer's comments, The section (Section 3.2 in original manuscript) “Given that strain types and production processes can affect fermentation and then change the composition of aroma components.…VOCs of NMBB had great changes in esters, pyrazines, ketones, acids, aldehydes, alcohols, amines, and aromatics, and according to sensory scores, these changes were developed in a positive direction. These change indicated that co-fermentation could effectively improve flavor of natto.” has been modified as follows: Given that strain types and production processes can affect fermentation and then change the composition of aroma components. SS, SFB, NMB, and NMBB were identified by HS-SPME-GC-MS. As shown in Table S1, 133 kinds of volatile compounds contents were identified. It was found that the increase of TA content and the decrease of pH in the sample did not increase the content of volatile acids. The reason was maybe that the increased acid is mainly non-volatile acid. To further examine the contribution of these 133 volatile compounds to the overall aroma profile of natto, their OAVs were calculated according to their concentrations and odor thresholds (Table S2).

As shown in Table 1, volatile substances with OAV > 10 in Table S2 are listed separately in order to more clearly find the substances that contribute significantly to the overall flavor of the sample. Aldehydes are mainly derived from lipid oxidation and Strecker amino acid degradation [6]. Precisely, linear fatty aldehydes and alkene aldehydes usually originate from lipid oxidation. Decanal, nonanal, hexanal and octanal originate from the oxidation of unsaturated fatty acids. In contrast, branched-chain aldehydes, such as benzaldehyde, 3-methylbutyraldehyde, and phenylacetaldehyde, are generated by the degradation of Strecker amino acids [7]. Due to the low flavor threshold, aldehydes contribute significantly to the unique flavor of natto products. Ketones can be generated by amino acids or by the Maillard reaction [8]. Alcohols may be produced from the metabolism of unsaturated fatty acids of microorganism [9]. Pyrazines can be generated by the Maillard reaction or the thermal reaction of microbial metabolites [10]. Acids are the lipid hydrolysate of soybeans, and have a sour or oily odor [11]. Organic acid volatiles may also be another cause of the pungent smell of natto. In the presented study, most of amines have a stimulating taste, which was the main reason for the ammonia flavor of natto. N,N-dimethyl-methylamine, formamide, 3-methyl-butanamide, ethosuximide were detected in NMB, but not in NMBB. Natto fermented by GUTU09 produces an ammonia smell, which is consistent with the report from Chen, Wang, Hu, Qian, Yao, Wang and Zhang [1]. The disappearance of amines indicated that ammonia taste of natto was effectively reduced. The volatile substances of NMBB mainly included alcohols (G) and ketones (D) aldehydes (F). Trimethyl-pyrazine (nutty, cocoa, and roasted flavors), 2,3-butanedione (sweet cream), decanal (sweet and flowery), 1-octen-3-ol (mushroom) were the main volatile substances in NMBB. NMB mainly included esters (B) and ketones (D) aldehydes (F). Liu, Su and Song [8] found that most esters were described as having fruit and floral flavors, such as pineapple, nut, banana, and apple flavors. It is speculated that acids and alcohols can be synthesized a large number of esters via esterification [12]. Bifidobacterium was added to fermented soybean, which resulted in significant changes in the important characteristics and flavor of natto. The reduction of volatile acids was consistent with the changes of FAA content, which may be converted into other organic compounds. After the addition of BZ25, volatile amine disappeared, volatile aldehyde and ketone species decreased, and alcohols increased. Evidently, the volatile components of natto were usually related to microbial metabolism [13]. Compared with the sensory evaluation results, it is the disappearance of some irritating volatile substances that makes the NMBB sample more popular. Volatile substances of NMBB had great changes in esters, pyrazines, ketones, acids, aldehydes, alcohols, amines, and aromatics, and according to sensory scores, these changes were developed in a positive direction. These changes indicated that co-fermentation could effectively improve the flavor of natto. He, et al. [13] found that the co-fermentation of three strains during natto fermentation may facilitate the formation of flavors.

 3.3.1 Principal component analysis (PCA) and Hierarchical cluster analysis (HCA)

To visualize a total picture of the distributions of 73 odor-active compounds (OAV > 1) in all samples, PCA was applied (Fig. 2A). PCA showed the major differences among the volatile profiles of four samples. The first principal component (PC1) of the PCA accounted for 50.1 %, and PC2 explained an additional 29.7 % of the total variability. And the cumulative contribution rate reached 79.8%. PCA Score Plot (Fig. 2A) shows that NMB and SFB were located in the first and second quadrants and positively correlates with PC2. SS and NMBB were located in the third and fourth quadrants and negatively correlate with PC2. Four samples were distributed in different quadrants, indicating that there were significant differences in flavor substances between samples. The characteristic components of SS were 2-methyl-propanoic acid ethyl ester (B4), 3-octanone (D8), 3-methyl-butanal (F3), octanal  (F5), phenylacetaldehyde (F6), eucalyptol (G20), 2-pentyl-furan(I1), 2-ethyl-furan (I3), those of SFB were 1-hexanol (G4) and 2-methoxy-phenol(H1), thoseof NMB was N,N-dimethyl-Methylamine (A1),  2-Ethyl-3,5-dimethyl-pyrazine (C3), acetic acid (F1),  2-ethyl-butanoic acid (F2) and 2-methyl-butanal (G4), and those of NMBB were hexanoic acid ethyl ester (B1), trimethyl-pyrazine (C1) 2-hexadecanone (D3), 2,3-butanedione (D4), nonanal (F2), 3-octanol (G3), biphenyl (H2) and naphthalene (J6). Therefore, the characteristic flavor components of natto can be changed by mixed fermentation of two bacteria.

Fig. 2(B) shows that hierarchical clustering and heat map visualization of volatile compounds of four samples in different treatments (OAV > 10). The gallery plot of four samples was applied to visualize the differences of aroma compounds. The aroma components of natto produced by co-fermentation were different from those produced by single strain GUTU09. Next, we used clustering analysis for Volatile compounds content and found that the four groups could be divided into three categories. Class I (SFB and NMB) included high content of acetic acid (E1), 2-ethyl-3,5-dimethyl-pyrazine (C3), 2-methyl-butanal (F4), 1-hexanol (G4), 2-methyl-3-hexanol (G9) and 2-methoxy-phenol (H1). Class II (SS) contained high content of 2-pentyl-furan (I1), 2-ethyl-furan (I3), hexanal (F9), 6-methyl-5-hepten-2-one (D19), eucalyptol (G20), 3-methyl-butanal (F3), 2-mmethyl-propanoic acid ethyl ester (B4), octanal (F5), 3-octanone (D8) and phenylacetaldehyde (F6). Class III (NMBB) mainly included high content of 2,3-hexadecanone (D3), 2,5-dimethyl-pyrazine (C2), trimethyl-pyrazine (C1), hexanoic acid ethyl ester (B1), 1-octen-3-ol (G1), styrene (J3), biphenyl (H2), 3-octanol (G3), naphthalene (J6), 2,3-butanedione (D4) and 2,3-dihydro-benzofuran (I2). Fig. 2(B) shows that the volatile substances of NMBB were accumulated more than NMB, and the flavor substances that contribute more to the whole are very different from NMB. PCA plot and gallery map show that mixed fermentation can greatly change the composition of volatile compounds of natto, thus affecting the sensory score and increasing people's acceptance of natto (lines 316-442).

Round 2

Reviewer 1 Report

The revised manuscript was improved significantly according to the reviewers' comments.

However, the manuscript still has need more critical discussion of the results. Moreover, moderate English changes are required especially for the newly added paragraphs. 

For example, lines between 305-308  repeat the synergistic effect but without any justification or citation. 

lines between 319-321 "It was found that the increase of TA content and the decrease of pH in the sample did not increase the content of volatile acids. "

It was found by who? 

" The reason was maybe that the increased acid is mainly non-volatile acid"

What do you mean?

So, I suggest revising carefully the Results and discussion section. It is possible that some mistakes occurred due to corrections that the authors operated in the text. 

Author Response

A detailed list (point to point) of responses to each item of the review comments

Q: The manuscript still has need more critical discussion of the results. Moreover, moderate English changes are required especially for the newly added paragraphs.

A: Thanks. We quite agree with the reviewer's comments and modified the section. The section (lines 325-363 - in original manuscript) “Given that strain types and production processes can affect fermentation and then change the composition of aroma components. SS, SFB, NMB, and NMBB were identified by HS-SPME-GC-MS. As shown in Table S1, 133 kinds of volatile compounds contents were identified. It was found that the increase of TA content and the decrease of pH in the sample did not increase the content of volatile acids. The reason was maybe that the increased acid is mainly non-volatile acid. To further examine the contribution of these 133 volatile compounds to the overall aroma profile of natto, their OAVs were calculated according to their concentrations and odor thresholds (Table S2).

As shown in Table 1, volatile substances with OAV > 10 in Table S2 are listed separately in order to more clearly find the substances that contribute significantly to the overall flavor of the sample. The volatile substances of NMBB mainly included alcohols (G) and ketones (D) aldehydes (F). Trimethyl-pyrazine (nutty, cocoa, and roasted flavors), 2,3-butanedione (sweet cream), decanal (sweet and flowery), 1-octen-3-ol (mushroom) were the main volatile substances in NMBB. Aldehydes are mainly derived from lipid oxidation and Strecker amino acid degradation [34]. Precisely, linear fatty aldehydes and alkene aldehydes usually originate from lipid oxidation. Decanal, nonanal, hexanal and octanal originate from the oxidation of unsaturated fatty acids. In contrast, branched-chain aldehydes, such as benzaldehyde, 3-methylbutyraldehyde, and phenylacetaldehyde, are generated by the degradation of Strecker amino acids [35]. Due to the low flavor threshold, aldehydes contribute significantly to the unique flavor of natto products. Ketones can be generated by amino acids or by the Maillard reaction [1]. Alcohols may be produced from the metabolism of unsaturated fatty acids of microorganism [36]. The volatile substances of NMB mainly included esters (B) and ketones (D) aldehydes (F). Liu, Su and Song [1] found that most esters were described as having fruit and floral flavors, such as pineapple, nut, banana, and apple flavors. It is speculated that acids and alcohols can be synthesized a large number of esters via esterification [37]. Most of amines have a stimulating taste, which was the main reason for the ammonia flavor of natto. N,N-dimethyl-methylamine, formamide, 3-methyl-butanamide, ethosuximide were detected in NMB, but not in NMBB. Natto fermented by single bacterium produces an ammonia smell, which is consistent with the report from Chen, et al. [38]. The disappearance of amines indicated that ammonia taste of natto was effectively reduced. After the addition of BZ25, volatile amine disappeared, volatile aldehyde and ketone species decreased, and volatile alcohol increased. Evidently, the volatile components of natto were usually related to microbial metabolism [39]. Compared with the sensory evaluation results, it is the disappearance of some irritating volatile substances that makes the NMBB sample more popular. Volatile substances of NMBB had great changes in esters, pyrazines, ketones, acids, aldehydes, alcohols, amines, and aromatics, and according to sensory scores, these changes were developed in a positive direction. These changes indicated that co-fermentation could effectively improve the flavor of natto. He, et al. [39] found that the co-fermentation of three strains during natto fermentation may facilitate the formation of flavors” has been modified as follows: Given that strain types and production processes can affect fermentation and then change the composition of aroma components. SS, SFB, NMB, and NMBB were identified by HS-SPME-GC-MS. As shown in Table S1, 133 kinds of VOCs were identified. Pyrazines, ketones and acids are abundant in NMB. The main VOCs in NMBB were pyrazines, ketones, acids, and alcohols. The difference between the main flavor substance in the NMB and the NMBB was alcohols, which was caused by the addition of BZ25. Compared with similar reports, it is found that the main VOCs of natto are the same but also different, which reflects both commonness and individuality. For example, Kaczmarska, et al. [35] found the volatile profile of the natto was dominated by alkylpyrazines and ketones. Chen, et al. [7] found that the main VOCs in natto were esters, ketones, pyrazines, and phenols. The different strains and fermentation conditions lead to the difference in natto volatiles components. To further examine the contribution of these 133 volatile compounds to the overall aroma profile of natto, their OAVs were calculated according to their concentrations and odor thresholds (Table S2).

As shown in Table 1, VOCs with OAV > 10 in Table S2 are listed separately to more clearly find the substances that contribute significantly to the overall flavor of the sample. Trimethyl-pyrazine, 2,3-butanedione, decanal, nonanal, 1-octen-3-ol, acetic acid, and 2-methoxy-phenol were the main VOCs in NMBB. Pyrazines were an important type of VOCs in natto. Pyrazines are formed by Bacillus subtilis or natto bacteria [36]. Liu, et al. [1] found that the key VOCs in natto were 2,3-butanedione, 5-methyl-2-hexanone, furaldehyde, acetic acid, and 2,5-dimethylpyrazine etc. Ketones can be generated by amino acids or by the Maillard reaction [1]. Kimura and Kubo [37] found that the key VOCs in natto were 2,3-butanediol, geranylacetone, isobutyl acetate, isovarelic acid, 2,5-dimethylpyrazine, and trimethylpyrazine etc. The key VOCs in the NMB were benzeneacetic acid methyl ester, 2-ethyl-3,5-dimethyl-pyrazine, 2-heptanone, acetic acid, 2-ethyl-butanoic acid, 2-methyl-butanal, 2-ethyl-furan, and 2-methoxy-phenol. According to the VOCs of NMB, NMBB, and the natto reported in these articles, it was found that pyrazines were the main VOCs of these natto, which indicated that Bacillus subtilis or natto bacteria could synthesize pyrazines during the fermentation of natto. The differences in VOCs also reflect the personality differences. Most amines have a stimulating taste, which was the main reason for the ammonia flavor of natto. N,N-dimethyl-methylamine, formamide, 3-methyl-butanamide, and ethosuximide were detected in NMB, but not in NMBB. Natto fermented by a single bacterium produces an ammonia smell, which is consistent with the report from Chen, et al. [38]. The disappearance of volatile amine indicated that the ammonia taste of natto was effectively reduced. After the addition of BZ25, volatile amine disappeared, volatile aldehydes and ketones species decreased, and volatile alcohol increased. Alcohols may be produced from the metabolism of unsaturated fatty acids of microorganisms [39]. The VOCs of natto were usually related to microbial metabolism [40]. Compared with the sensory evaluation results, it is the disappearance of some irritating VOCs that makes the NMBB sample more popular. Sadineni, et al. [41] found that the co-fermentation of two strains during mango wine fermentation may decrease volatile acid and improve flavor. (Lines 325 – 368)

Q: lines between 305-308 repeat the synergistic effect but without any justification or citation.

A: Thanks. We have substantially modified the section. The section (lines 305-308 - in original manuscript) “Results showed that the two strains have a synergistic effect on acid production, which produced synergistic effect and promoted the acid production of BZ25.” has been modified as follows: Results showed that the two strains have a synergistic effect on acid production, which promoted the acid production of BZ25. Bujna, et al. [33] found that more acids were produced by co-fermentation of Lactobacillus and Bifidobacterium. (Lines 315 – 318)

Q: lines between 319-321 "It was found that the increase of TA content and the decrease of pH in the sample did not increase the content of volatile acids."It was found by who? "The reason was maybe that the increased acid is mainly non-volatile acid "What do you mean?

A: Thanks. We have modified it. The section (lines 319-321 - in original manuscript) “It was found that the increase of TA content and the decrease of pH in the sample did not increase the content of volatile acids. The reason was maybe that the increased acid is mainly non-volatile acid” was deleted. The section “Pyrazines, ketones and acids are abundant in NMB. The main volatile substances in NMBB were pyrazines, ketones, acids, and alcohols. The difference between the main flavor substance in the NMB and the NMBB was alcohols, which was caused by the addition of BZ25. Compared with similar reports, it is found that the main VOCs of natto are the same but also different, which reflects both commonness and individuality. For example, Kaczmarska, et al. [35] found the volatile profile of the natto was dominated by alkylpyrazines and ketones. Chen, et al. [7] found that the main VOCs in natto were esters, ketones, pyrazines, and phenols. The different strains and fermentation conditions lead to the difference in natto volatiles components.” was added. (Lines 329 -338)
